# Pharmacologic hyperstabilisation of the HIV-1 capsid lattice induces capsid failure

KM Rifat Faysal[1†], James C Walsh[1†], Nadine Renner[2‡], Chantal L Márquez[1‡], Vaibhav B Shah[1], Andrew J Tuckwell[1], Michelle P Christie[3], Michael W Parker[3,4], Stuart G Turville[5], Greg J Towers[6], Leo C James[2], David A Jacques[1*], Till Böcking[1*]

[1]EMBL Australia Node in Single Molecule Science, School of Biomedical Sciences, UNSW, Sydney, Australia; [2]MRC Laboratory of Molecular Biology, Cambridge, United Kingdom; [3]Department of Biochemistry and Pharmacology, Bio21 Molecular Science and Biotechnology Institute, University of Melbourne, Melbourne, Australia; [4]Structural Biology Unit, St. Vincent's Institute of Medical Research, Fitzroy, Australia; [5]The Kirby Institute, UNSW, Sydney, Australia; [6]Division of Infection and Immunity, University College London, London, United Kingdom

**Abstract** The HIV-1 capsid has emerged as a tractable target for antiretroviral therapy. Lenacapavir, developed by Gilead Sciences, is the first capsid-targeting drug approved for medical use. Here, we investigate the effect of lenacapavir on HIV capsid stability and uncoating. We employ a single particle approach that simultaneously measures capsid content release and lattice persistence. We demonstrate that lenacapavir's potent antiviral activity is predominantly due to lethal hyperstabilisation of the capsid lattice and resultant loss of compartmentalisation. This study highlights that disrupting capsid metastability is a powerful strategy for the development of novel antivirals.

## Editor's evaluation

This important study substantially advances our understanding of the effects of small molecule inhibitors on the structural integrity and stability of the HIV-1 capsid. Rigorous biochemical assays and state-of-the-art microscopy provide compelling support for the conclusions. The work will be of broad interest.

*For correspondence:
d.jacques@unsw.edu.au (DAJ);
till.boecking@unsw.edu.au (TB)

[†]These authors contributed equally to this work
[‡]These authors also contributed equally to this work

Competing interest: The authors declare that no competing interests exist.

## Introduction

The cell is a hostile environment for HIV, as the reverse transcribed cDNA genome is a target for innate immune sensors which unleash a potent interferon response that can suppress replication (*Zuliani Alvarez et al., 2022*; *Lahaye et al., 2013*; *Rasaiyaah et al., 2013*). For a productive infection to occur, the reverse transcribing genome must be trafficked through the cytoplasm, enter the nucleus and integrate into the preferred sites in the host chromatin, all while evading detection by the host cell. The viral capsid facilitates these early steps in the replication cycle by encapsulating the genome and associated viral enzymes. In doing so, it protects the genome from being sensed and destroyed by nucleases, prevents loss of viral enzymes from the reverse transcription complex, and forms the interface through which all cytoplasmic, and many nuclear, host-virus interactions occur.

The conical capsid shell is comprised of ~1500 copies of the capsid protein (CA), which spontaneously assemble into a lattice. This lattice consists of mostly hexamers and exactly 12 pentamers to form a closed fullerene cone (*Ganser et al., 1999*; *Pornillos et al., 2011*). While the capsid must be stable enough to transit the cytoplasmic compartment without exposing the genome, it must also be able to release the reverse transcribed cDNA at the appropriate time and in the appropriate location

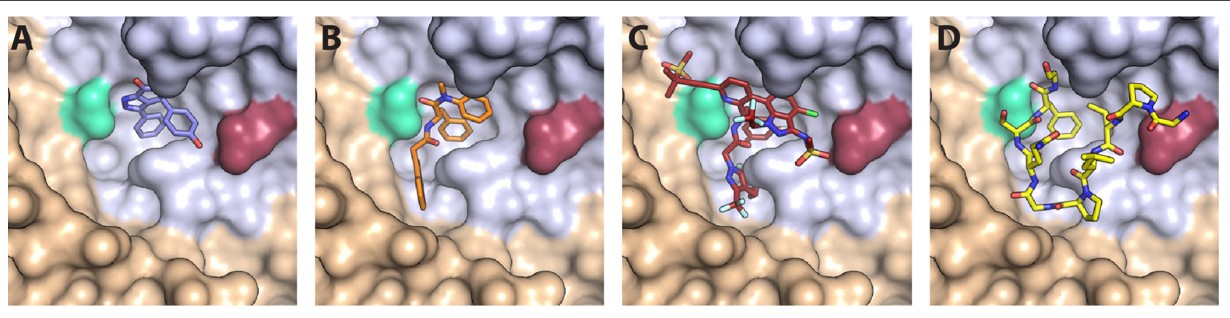

**Figure 1.** CA hexamer structures in complex with FG-binding pocket ligands. Two neighbouring CA molecules (grey, light brown) are shown as surface representation. Landmark residues N57 (pale green) and N74 (red-brown) are highlighted. Ligands are shown as sticks: (**A**) BI-2. (**B**) PF74. (**C**) LEN. (**D**) CPSF6 peptide. PDB IDs in **A**, **B** and **D**: 4U0F, 4U0E, 4U0A (*Price et al., 2014*). PDB ID in **C**: 6V2F (*Link et al., 2020*). Images were generated with PyMol version 2.3.5.

in the nucleus. This process is called capsid uncoating and its regulation and mechanism are poorly understood. Recently, we demonstrated that uncoating proceeds through two discrete steps in vitro: capsids opening, in which the integrity of the cone is compromised and encapsidated proteins are released; and catastrophic disassembly of the lattice by release of CA (*Márquez et al., 2018*). We have also shown that recruitment of cellular cofactors or binding of pharmacological agents to the capsid can greatly alter both processes.

To engage with host cofactors, the mature HIV capsid utilises at least three cytoplasm-facing surfaces (*Temple et al., 2020*). One of these is the central pore, which is formed by a ring of positively charged arginine residues (R18 ring) at the sixfold symmetry axis in CA hexamers and at the fivefold axis in pentamers (*Jacques et al., 2016*). It serves to recruit nucleotides required for reverse transcription and as a binding site for inositol hexakisphosphate (IP6), a metabolite present at concentrations of ~40–50 μM in human cells (*Bunce et al., 1993*; *Letcher et al., 2008*; *Veiga et al., 2006*). IP6 is specifically packaged into immature virions during assembly in producer cells, leading to a ≥10–fold enrichment, and its interaction with the central pore is essential for the assembly and stability of the capsid (*Dick et al., 2018*; *Mallery et al., 2018*; *Obr et al., 2021*; *Sowd and Aiken, 2021*). The central pore has also been implicated in the capsid's interaction with microtubule-based motor proteins (*Huang et al., 2019*).

Another important interface is the cyclophilin binding loop. As its name suggests, this largely unstructured loop protrudes from the capsid surface to recruit cyclophilin A (CypA). This interaction is mediated by a conserved glycine-proline motif (G89-P90) that inserts into the CypA active site. The implications of this interaction are still to be fully understood, but are thought to include viral evasion of innate host defences (*Kim et al., 2019*; *Miles et al., 2020*; *Rasaiyaah et al., 2013*; *Towers et al., 2003*). Importantly, we have previously shown that this interaction can be exploited in vitro to 'paint' the capsid (*Márquez et al., 2018*). By using a fluorescently labelled CypA, we found that it is possible to detect, quantify, and monitor disassembly of the CA lattice in permeabilised virions by total internal reflection fluorescence (TIRF) microscopy without significantly influencing the capsid opening process.

The third host-interaction surface, the FG-binding site, is a hydrophobic pocket in the CA N-terminal domain situated near the intra-hexameric junction between CA molecules (*Figure 1*). This site serves as the binding interface for several host factors, including the cytoplasmic protein Sec24C (*Rebensburg et al., 2021*), components of the nuclear pore complex including NUP153 (*Matreyek et al., 2013*) and other permeability barrier-forming nucleoporins (*Dickson et al., 2024*; *Fu et al., 2024*), and the nuclear protein CPSF6 (*Bhattacharya et al., 2014*; *Price et al., 2014*). Sequential interactions with these proteins are crucial for nuclear entry and correct integration site targeting (*Bejarano et al., 2019*; *Saito et al., 2016*; *Schaller et al., 2011*). Each of these proteins interacts with the capsid via a phenylalanine-glycine (FG) motif. Additionally, several generations of antiviral compounds also target this site by effectively mimicking the FG-motif (*Price et al., 2014*). Interestingly, the potency and mechanism of action of these agents varies, despite their shared binding site.

Compounds that target the FG-binding pocket include BI-2 (*Lamorte et al., 2013*), PF74 (*Blair et al., 2010*), and lenacapavir (LEN, GS-6207) (*Link et al., 2020*). BI-2 has a relatively small binding

footprint on the CA N-terminal domain, barely extending beyond the FG-binding pocket (*Figure 1A*). This limited interaction is reflected in the relatively weak binding affinity ($K_D$ = 1.2 μM) and high drug concentration required to inhibit infection by 50% during the early stage of infection (half maximal inhibitory concentration, IC50=3 μM) (*Price et al., 2014*). PF74, on the other hand, extends its interaction beyond the FG-binding pocket to make additional contacts with a neighbouring CA molecule (*Figure 1B*). These bridging contacts are thought to be responsible for PF74's 10-fold tighter $K_D$ (120 nM) and IC50 (relative to BI-2). Both BI-2 and PF74 can compete with host cofactor binding (Sec24C, Nup153, and CPSF6) (*Fricke et al., 2014*; *Matreyek et al., 2013*; *Peng et al., 2014*; *Price et al., 2014*; *Rebensburg et al., 2021*), and also directly affect capsid structure and stability. While both compounds have capsid-opening activity (*Bhattacharya et al., 2014*; *Fricke et al., 2014*; *Shi et al., 2011*), leading to a block of reverse transcription (*Jennings et al., 2020*; *Mallery et al., 2018*; *Sowd et al., 2021*), PF74 also increases the stiffness of the capsid (*Rankovic et al., 2018*) and stabilises hexameric CA lattices (*Bhattacharya et al., 2014*; *Mallery et al., 2018*).

In contrast to BI-2 and PF74, which have found use exclusively in the laboratory, LEN is a first-in-class HIV-1 capsid inhibitor in clinical trials and approved by the European Union and by the U.S. Food and Drug Administration for clinical use in patients with multidrug-resistant HIV-1 infection (*Dvory-Sobol et al., 2022*; *Patel et al., 2023*). Structurally, LEN makes extensive contacts across two neighbouring CA monomers (*Figure 1C*), allowing for a very high-affinity interaction ($K_D$ = 215 pM). It interferes with early and late phases of the HIV-1 replication cycle at low and high pM concentrations, respectively, making it orders of magnitude more potent than PF74 and BI-2 (*Bester et al., 2020*; *Link et al., 2020*). Successive post-entry steps differ in their sensitivity to the drug (*Bester et al., 2020*; *Link et al., 2020*), whereby integration of HIV-1 DNA into host chromatin is inhibited most potently (<500 pM), followed by nuclear import and reverse transcription, which is inhibited at high pM (*Sowd et al., 2021*) to low nM LEN (*Bester et al., 2020*). Remarkably, imaging studies (*Bester et al., 2020*) and biochemical assays (*Selyutina et al., 2022*) showed that LEN increased the number of viral cores in the cytoplasm in a dose-dependent manner. Thus, the drug exhibited apparently contrasting inhibitory effects at higher concentrations: inhibition of reverse transcription and stabilisation of viral cores.

Here, we have used single-molecule fluorescence imaging to show that LEN breaks IP6-stabilised capsids while preventing disassembly of CA lattices with open edges. Indirect visualisation of LEN binding to authentic capsids in real time reveals that capsid breakage depends on the occupancy of FG binding pockets. In vitro assembly assays show that LEN and IP6 synergise to accelerate CA assembly but promote tubular (LEN) versus conical (IP6) structures, leading to the formation of aberrant capsid structures. Altogether, our data suggest that substoichiometric binding of LEN stabilises the hexameric CA lattice at the cost of fully closed capsids, leading to loss of reverse transcription. This mechanism becomes dominant in the high pM range, close to the inhibitory concentration that leads to 95% inhibition (IC95) of viral replication.

## Results

### Single-virion analysis of intrinsic capsid stability and uncoating

We used a single-molecule fluorescence imaging assay to measure the intrinsic capsid stability and uncoating kinetics at the level of individual viral particles (*Márquez et al., 2018*) and then measured the effect of LEN treatment on these processes. As shown schematically in *Figure 2A*, we used pore-forming proteins to permeabilise GFP-loaded HIV particles immobilised at the bottom surface of a microfluidic channel device. Using TIRF microscopy, we then detected the stepwise loss of the GFP signal for each virion appearing as a bright diffraction-limited spot in the field of view. Upon membrane permeabilisation, virions with a 'leaky' capsid lost their entire GFP signal in a single step (*Figure 2B*). In contrast, virions containing an intact capsid retained the pool of GFP inside the capsid. This residual (low intensity) GFP signal was lost in a second step upon spontaneous loss of capsid integrity (*Figure 2B*, 'opening'), whereby the lifetime of each opening capsid was given by the time difference between the two GFP release steps. A subset of GFP-containing capsids remained at the end of the experiment (*Figure 2B*, 'closed'), because the 30-min imaging period was not long enough to observe opening of all capsids. Finally, virus preparations also contained a subset of immature particles that remain at the initial high intensity level throughout the imaging period because the GFP

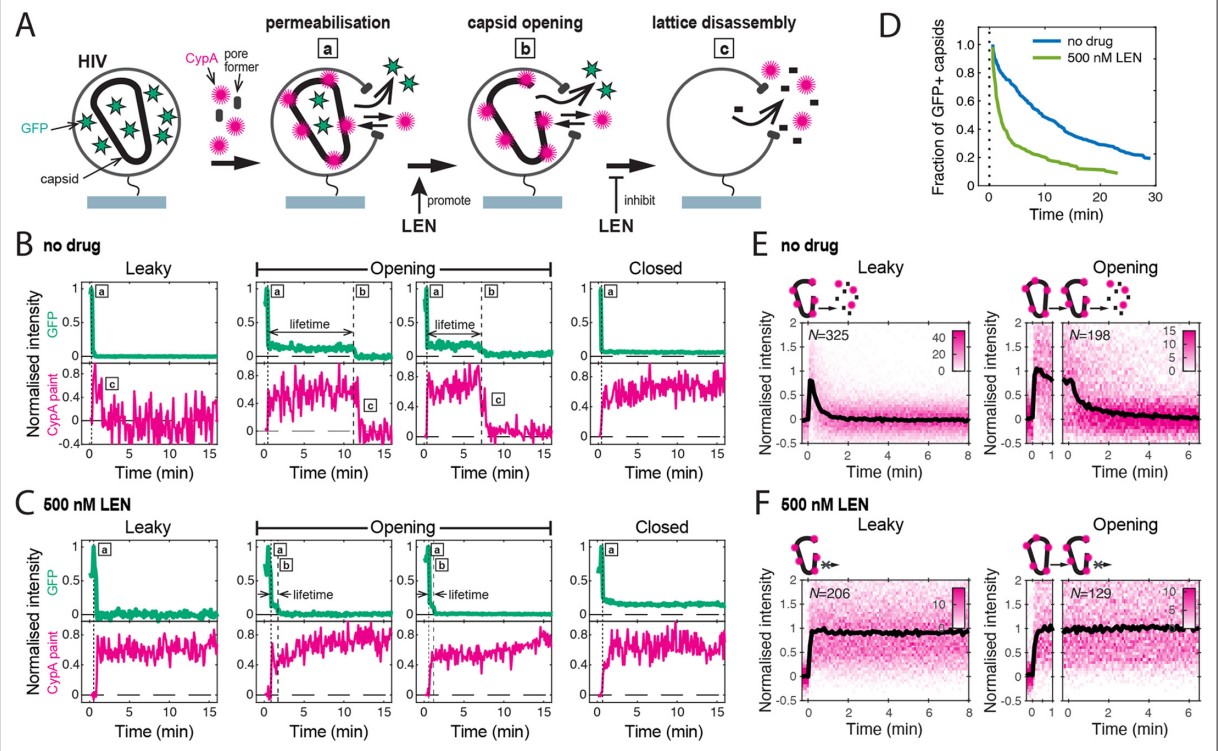

**Figure 2.** Single-molecule HIV capsid uncoating kinetics measured by TIRF microscopy. (**A**) Schematic diagram of a viral particle at different stages of uncoating detected in the assay. HIV particles were produced using a proviral construct with internal GFP that is released from the viral Gag protein during maturation and packaged as a solution phase marker inside the two compartments bound by the viral membrane and the capsid, respectively. These GFP-loaded HIV particles are immobilised on the coverslip surface and permeabilised in the presence of AF568-labelled CypA while recording fluorescence traces at the locations of individual HIV particles by TIRF microscopy. Permeabilisation of the viral membrane (step a) with a pore-forming protein leads to loss of ~80–90% of the GFP signal corresponding to the pool of GFP outside the capsid. AF568-CypA molecules diffuse through the membrane pores and bind to the capsid to reach a level that is proportional to the number of CA subunits in the capsid. Capsid opening (step b) leads to loss of the residual GFP that is inside the capsid. CA lattice disassembly (step c) is apparent from the rapid loss of the CypA paint signal. (**B, C**) Example GFP release (blue-green) and CypA paint (magenta) traces for particles with capsids that are already leaky (i.e. contain defects and release all GFP in one step), undergo opening at various times after permeabilisation or remain closed throughout the observation period. In the absence of drug (**B**), the CypA paint intensity decays rapidly when the capsid is no longer closed (complete loss of GFP signal). In the presence of 500 nM LEN (**C**), the CypA paint signal remains constant even when the GFP signal is completely lost showing that the drug stabilises the ruptured capsid. (**D**) Analysis of the capsid lifetimes from all single-molecule GFP release traces in the field of view to yield capsid survival curves (including 'opening' and 'closed', excluding 'leaky'). The faster decay in the presence of 500 nM LEN compared to no drug control shows that LEN induces rupture of the capsid. Data from a representative experiment (total number of traces): no drug (615); 500 nM LEN (281). (**E, F**) Analysis of all single-molecule CypA paint traces to yield heatmaps (magenta) and median traces (black line) of the CypA intensity measured at particles with leaky (left) or opening (right) capsids in the absence (**E**) or presence of 500 nM LEN (**F**). LEN prevents dissociation of CA from the lattice of capsids that are no longer closed cones. The number of HIV particles (N) for each condition is specified in the top left corner of the corresponding heatmap.

The online version of this article includes the following figure supplement(s) for figure 2:

**Figure supplement 1.** Maturation of HIV produced with Gag-iGFP.

**Figure supplement 2.** Interpretation of TIRF uncoating data and limitations of the assay.

**Figure supplement 3.** The pore-forming protein SLO does not affect capsid stability.

remains bound to the viral membrane as part of the immature Gag lattice (*Figure 2—figure supplement 1*).

We used step-fitting to classify the typically 300–1,000 virions per field of view according to the GFP release profiles defined above. Immature virions (6.9 ± 3.1% of all particles) were excluded from further analysis. Most of the mature virions showed leaky release profiles (56 ± 5%), which we attribute to incompletely assembled capsids that are also seen in cryoelectron tomography reconstructions of mature HIV (*Mattei et al., 2016*). Capsids with opening and closed intensity profiles (33.7 ± 5.0% and 10.4 ± 3.5%, respectively) were further analysed to quantify the kinetics of capsid opening.

We obtained a capsid survival curve (lifetime distribution) by plotting the fraction of capsids that remain intact (GFP-positive) as a function of time. The survival curve for untreated virions (*Figure 2D*, no drug) showed a biphasic decay profile with ~30% short-lived (half-life of 1.0±0.5 min) and ~70% long-lived (half-life of 15±3 min) capsids, consistent with previous observations (*Mallery et al., 2021*; *Márquez et al., 2018*; *Renner et al., 2023*). Long-lived capsids can be stabilised by the host cofactor IP6 (see below) and are considered functionally relevant (see *Figure 2—figure supplement 2* for a detailed discussion of the capsid stability types observed in this assay). Control experiments showed that the fraction of IP6-stabilised capsids was independent of the concentration of the pore-forming protein used to permeabilise the viral membrane, confirming that it does not affect capsid stability (*Figure 2—figure supplement 3*).

As a complementary measurement, we used AF568-labelled CypA as a 'paint' that binds transiently to the outside of the capsid, rapidly reaching a dynamic equilibrium, whereby the AF568-CypA intensity is proportional to the number of CA subunits in the lattice. Importantly, AF568-CypA was used at concentrations (0.5–1 µM) where fewer than 4% of the available cyclophilin loops are occupied, and we have previously shown that uncoating kinetics are not affected under these conditions (*Márquez et al., 2018*). Single-molecule analysis showed that the AF568-CypA signal remained constant while the capsid was intact (*Figure 2B*, 'closed') but decayed to background levels after the capsid opened (*Figure 2B*, 'leaky' and 'opening'). Analysis of all leaky and opening traces aligned to the time of capsid opening showed that the median CypA signal decayed with a half-life of less than 1 min (*Figure 2E*). This rapid decay is consistent with a failure cascade that propagates across the whole capsid resulting in complete lattice disassembly. Taken together, our single-molecule analysis shows that GFP release pinpoints the time the first defect appears in the capsid while the CypA paint signal provides an indirect read-out for the disassembly kinetics of the CA lattice thereafter.

## LEN induces capsid opening but prevents loss of CA from the lattice of open capsids

In the next set of experiments, we focused on the effect of LEN on the intrinsic capsid stability without capsid-binding cofactors; the interplay between IP6, which is essential for maintaining capsid stability in cells, and LEN is described in later sections. To measure the maximum effect of LEN on capsid uncoating, we added the drug at a concentration (500 nM) that leads to rapid binding to essentially all FG binding sites. Uncoating traces of single virions in the presence of 500 nM LEN (*Figure 2C*) revealed two fundamental differences to the single-molecule profiles described above. First, LEN treatment caused earlier release of the encapsidated GFP (*Figure 2C*, 'opening'), resulting in a faster decaying survival curve (*Figure 2D*, light green line) and a concomitant threefold decrease in the fraction of closed capsids at the end of the experiment. Second, the AF568-CypA signal of capsids that were defective to begin with (*Figure 2C*, 'leaky') or started to uncoat (*Figure 2C*, 'opening') remained constant. This striking stabilisation effect was also clear in the heatmaps of all leaky and opening capsids (*Figure 2F* and 500 nM LEN). Since the CypA paint traces of leaky and opening capsids showed the same characteristics, we combined these classes in subsequent analyses to assess the extent to which LEN could stabilise the lattice of open capsids. Taken together, the GFP release and CypA paint analysis in the presence of 500 nM LEN suggest that binding of LEN induces rupture of capsids (leading to early GFP release) but prevents the loss of CA subunits from defective or ruptured capsids (stable CypA paint signal).

To further quantify these apparently opposing effects on capsid integrity and CA lattice stability, we measured GFP release and CypA paint traces in the presence of LEN at concentrations ranging from 5 to 500 nM (*Figure 3*). The survival curves (*Figure 3A*) showed a pronounced concentration-dependent acceleration of capsid opening kinetics with an intermediate effect at 5 nM and the maximal effect at ≥50 nM LEN, leading to a two- to fourfold (5 nM and ≥50 nM LEN, respectively) decrease in the fraction of capsids that remained closed after 15 min (*Figure 3C*). Similarly, we observed a concentration-dependent increase in the fraction of particles with leaky capsids from 56 ± 5% (no drug) to 65 ± 7% (500 nM LEN) (*Figure 3—figure supplement 3*), which we attribute to rapid drug-induced capsid rupture that occurred before membrane permeabilisation.

Next, we analysed the CypA paint signal to determine the effect of LEN on the lattice after capsid integrity loss. LEN concentrations of ≥50 nM prevented disassembly of leaky and opening capsids, as evidenced by the stable CypA paint signal (*Figure 3D*), which persisted for at least 5 h

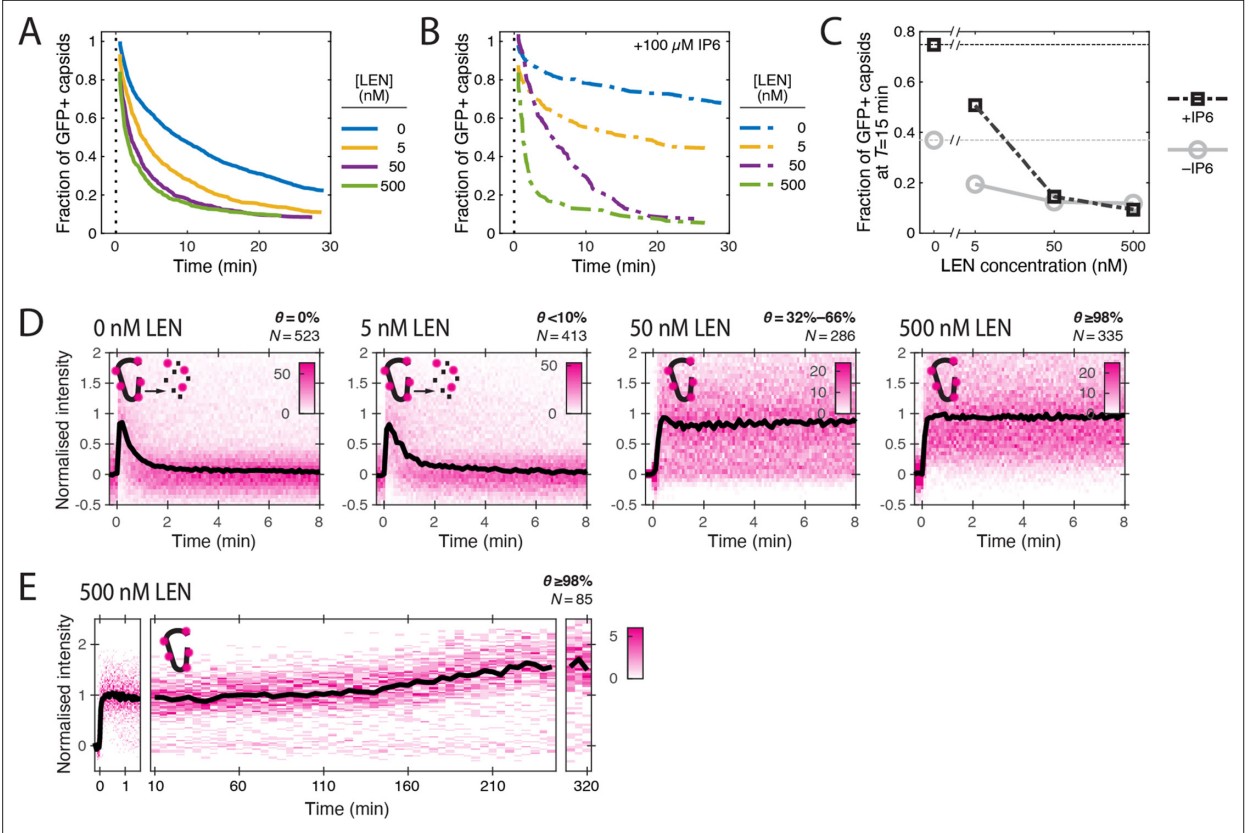

**Figure 3.** LEN accelerates capsid opening and subsequently prevents CA lattice disassembly. Single-molecule analysis of the effect of 0–500 nM LEN −/+100 µM IP6 on capsid uncoating via GFP release and CypA paint. (**A**) Capsid survival curves showing that the drug induces rupture of the capsid. Pooled data from multiple experiments (total number of traces/number of experiments): 0 nM (4325/10); 0.5 nM (1242/4); 5 nM (1585/4); 50 nM (1520/5); 500 nM (1048/4). (**B**) Capsid survival curves showing that IP6 inhibits capsid opening in the absence of LEN and partially counteracts the drug-induced rupture of the capsid at low but not high concentrations of LEN. Pooled data from multiple experiments (total number of traces/number of experiments): 0 nM LEN +IP6 (836/3); 5 nM LEN +IP6 (589/2); 50 nM LEN +IP6 (321/1); 500 nM LEN +IP6 (238/1). (**C**) Fraction of closed (GFP-positive) capsids at $t$=15 minutes of the uncoating experiments shown in **A** and **B**. (**D**) Heatmaps (magenta) and median traces (black line) of the CypA intensity measured at particles with leaky or opening capsids in the presence of 0–500 nM LEN showing that LEN stabilises the CA lattice of ruptured capsids above an occupancy ($\theta$) threshold of ~30–66%. The occupancy at the time of membrane permeabilisation was calculated as described in *Figure 4—figure supplement 1*. (**E**) Heatmap (magenta) and median trace (black line) of the CypA intensity of particles (leaky/opening) showing that 500 nM LEN prevents CA loss from the ruptured capsid for at least 5 hr. The number of HIV particles (N) for each condition in **D** and **E** is specified above the corresponding heatmap.

The online version of this article includes the following figure supplement(s) for figure 3:

**Figure supplement 1.** Heatmaps (magenta) and median traces (black line) of the CypA intensity measured at particles with leaky or opening capsids in the absence (**A**) or presence (**B**) of 100 µM IP6.

**Figure supplement 2.** Heatmaps (magenta) and median traces (black line) of the CypA intensity measured at particles with leaky (left) or opening (right) capsids.

**Figure supplement 3.** Effect of LEN on the fraction of particles that release their total GFP content in a single step.

(*Figure 3E*). Under the conditions used in the single-molecule uncoating assay, the capsid is exposed to the membrane-permeable drug for ~2 min before the pore forming protein permeabilises the viral membrane. Thus, we first calculated the occupancy at this time point to estimate the threshold required to prevent disassembly of defective capsids (see Materials and methods). Unsurprisingly, the low occupancy at 5 nM LEN (calculated to be <10%) was insufficient for lattice stabilisation. In contrast, 32–66% occupancy at 50 nM LEN was sufficient to prevent CA lattice disassembly. Based on these considerations, we estimate that 2–4 LEN molecules per hexamer are sufficient to prevent the release of CA subunits from a CA lattice with open edges.

In summary, LEN dose-dependently increased the proportion of open capsids but increased the stability of their lattices. The dose-dependence observed here differs from previous in vitro uncoating

measurements showing (partial) capsid lattice stabilisation down to 0.05 nM LEN; in those experiments 50–65% of cores stained with the irreversibly bound tetrameric probe CypA-dsRed remained detectable after 30 min of incubation (*Bester et al., 2020*). Nevertheless, our observation of capsids that do not lose CA despite having openings is consistent with the LEN dose-dependent increase in the number of CA-positive spots detected in the cytoplasm of infected cells (*Bester et al., 2020*; *Selyutina et al., 2022*).

## IP6 counteracts but does not prevent LEN-induced capsid rupture

We have previously shown that the cellular cofactor IP6 stabilises capsids and delays capsid opening in vitro (*Mallery et al., 2018*). Since IP6 is present in cells (typically 40–50 µM; *Bunce et al., 1993*; *Letcher et al., 2008*) and therefore expected to impact the effects of LEN in vivo, we asked if IP6 could prevent LEN-induced rupture of the capsid. As before, IP6 (100 µM) strongly stabilised capsids in permeabilised virions, leading to reduced capsid opening kinetics (*Figure 3B*) and a ~twofold increase in the fraction of closed capsids at 15 min compared to the control without IP6 (*Figure 3C*). IP6 partially counteracted the capsid-breaking effect at 5 nM LEN (compare *Figure 3A and B*), but even at this concentration the drug increased capsid opening kinetics relative to IP6 only (*Figure 3B*). At high LEN concentrations (≥50 nM), the survival curves measured in the presence and absence of IP6 were essentially the same, showing that IP6 was no longer able to slow the premature rupture of the capsid induced by the drug (*Figure 3B*). This acceleration in capsid opening led to an eightfold reduction in the fraction of closed capsids at 15 min compared to the control with IP6 only (*Figure 3B*). We conclude that IP6 partially protects capsids from LEN-induced structural damage, but only at low drug concentrations. The higher susceptibility to structural damage in the absence of IP6 is consistent with the observation that LEN more potently inhibits reverse transcription in vitro and in cells when IP6 levels are low (*Sowd et al., 2021*).

Finally, we asked if IP6 could act in conjunction with LEN to stabilise the CA lattice after capsid opening. As observed for other polyanions binding at the R18 ring at the centre of the CA hexamer (*Márquez et al., 2018*), IP6 by itself slows but does not prevent the catastrophic collapse of CA lattices with open edges (*Figure 3—figure supplement 1*). CA lattice dissociation of leaky and opening capsids is further slowed when IP6 is added to 5 nM LEN (<10% occupancy of FG sites at the time of membrane permeabilisation) (*Figure 3—figure supplement 2A and B*). At 50 nM LEN (>30% occupancy of FG sites), the CypA paint signal remains constant in the presence and absence of IP6 (*Figure 3—figure supplement 2C and D*), such that differences in stability during the time frame of the experiment are difficult to ascertain. Overall, these data suggest that IP6 further stabilises LEN-stabilised CA lattices but not to the extent where it can prevent CA release from lattices at low LEN occupancy.

## Slow LEN binding kinetics delay the structural drug effects at low concentrations

Next, we considered the kinetics of LEN binding to explain the observed differences in drug-induced capsid rupture between 5 nM and higher concentrations (50 and 500 nM). The LEN binding curves calculated using published association and dissociation rate constants (*Figure 4—figure supplement 1A*) predict that LEN occupancies ≥95% are reached within <2 min at 500 nM and ~15 min at 50 nM but reaching this level requires almost 3 hr at 5 nM LEN. Thus, we postulated that the intermediate capsid breakage kinetics observed at 5 nM LEN are due to slow binding kinetics such that occupancy levels required for fast capsid breakage are not reached quickly enough within the imaging period. To test this, we preincubated virions with 5 nM LEN for 4 hr, during which 95% of sites are occupied. Analysis of GFP release traces showed that preincubation with 5 nM LEN increased the extent of drug-induced capture rupture to that observed at 500 nM without preincubation (*Figure 4—figure supplement 1B*). Notably, >50% of the capsids had apparently already been ruptured inside the viral membrane, leading to an increase in 'leaky' traces relative to control (*Figure 4—figure supplement 1B*). This suggests that drug-induced capsid rupture manifests over time and that rupture, or build-up of the strain required for rupture, can already occur before release of the capsid from the virion. Finally, CypA paint analysis showed that CA lattice disassembly was inhibited with, but not without, preincubation (*Figure 4—figure supplement 1C*). Taken together, these observations suggest that

the full extent of the capsid-altering effects is observed after binding occupancy reaches the requisite level.

## Indirect fluorescence imaging of LEN binding kinetics to HIV capsids via displacement of a 'paint' probe for the FG pocket

To corroborate the LEN binding kinetics, we developed a new imaging assay to measure LEN binding to authentic capsids in real time by displacement of a fluorescent peptide derived from CPSF6 as a 'paint' probe that binds dynamically to the FG binding pocket, whereby the binding level is proportional to the fraction of sites that remain unoccupied with the tightly binding drug molecule. The requirement for a paint probe in this assay is that it has fast binding and dissociation kinetics such that it rapidly reaches a dynamic binding equilibrium to probe free sites without blocking them for drug binding. A CPSF6 peptide (CPSF6p, residues 313–327) with a C-terminal cysteine labelled with a fluorescent dye fulfilled these criteria, binding with a half-life of 11 s to the capsids of permeabilised virions on the TIRF microscope coverslip (*Figure 4—figure supplement 2*). Using single-molecule photobleaching for calibration of the fluorescence intensity, we determined the number of labelled CPSF6p bound at equilibrium per capsid at a range of concentrations (*Figure 4—figure supplement 2D*). At the CPSF6p paint probe concentration (200 nM) used in the LEN binding assay, ~40 labelled CPSF6p are bound at equilibrium per capsid, occupying <3% of FG binding pockets.

The design of the LEN binding assay is shown in *Figure 4A*. GFP-loaded HIV on the coverslip surface were permeabilised using a pore-forming protein and incubated for 2 hr in the presence of IP6 (30 µM) and labelled CPSF6p. During this period all capsids that cannot be stabilised by IP6 (leaky and short-lived) decayed away such that only long-lived IP6-stabilised capsids remained. These were identified as spots positive for GFP (as a capsid integrity marker) and CPSF6. We then added LEN to the flow channel and tracked the GFP and CPSF6p signals at the locations of individual virions over time. LEN led to a concentration-dependent decay of the CPSF6p intensity (*Figure 4B*) as LEN increasingly (and irreversibly on the time scale of this experiment) occupies FG binding pockets on the lattice. (We note that photobleaching does not contribute to the signal decay because the bound paint probes continually exchange with fresh molecules from solution). Thus, the decrease in CPSF6p intensity is an indirect read-out for LEN binding, and we converted this data into kinetic curves of LEN binding to authentic HIV capsids (*Figure 4C*). An independent repeat of this set of experiments is shown in *Figure 4—figure supplement 3A and B*. The association rate constant determined from the pooled data of both experiments ($k_{on}$ = 1.84 ± 0.78×10$^5$ M$^{-1}$ s$^{-1}$, *Figure 4D*) was within a factor of three of the published value ($k_{on}$ = 6.5 ± 0.3×10$^4$ M$^{-1}$ s$^{-1}$), while the $K_D$ of 0.51±0.58 nM was about two times the published value ($K_D$ = 0.24 ± 0.09 nM; *Link et al., 2020*). These differences are within experimental error of our method but could also reflect a difference in binding to an authentic capsid versus binding to a cross-linked CA hexamer used in the published work.

## LEN-induced capsid rupture kinetics depend on the occupancy of FG binding pockets

Next, we analysed the GFP release traces collected in the LEN-binding TIRF experiments described above (*Figure 4A*) to extract the capsid opening kinetics (*Figure 4F and G*). Importantly, the new assay design allows the functionally irrelevant subset of defective/IP6-insensitive capsids to disassemble before measuring the effect of LEN exclusively on the functionally relevant subset of IP6-stabilised capsids. This avoids the complexity of convolving intrinsic instability with drug-induced breakage. Remarkably, the stability of the IP6-responsive capsids in the absence of drug exceeded the level of IP6-mediated stabilisation we have previously observed, and we observed little or no loss of capsids over the 90-min imaging period (*Figure 4F*, 0 nM LEN). The same observation held when imaging was extended to 8 hr, and we estimate the half-life of IP6-stabilised capsids in this assay design to be on the order of day(s).

Survival analysis in the presence of 0.2–20 nM LEN (*Figure 4F*) revealed the following drug effects on capsid stability. Firstly, rupture of IP6-stabilised capsids accelerated with increasing LEN concentrations. This is apparent from the concentration-dependent faster decay of the survival curves and the corresponding decrease of the half-life of the intact capsid (*Figure 4G*, note the log-log scale) from 4.5 hr at 0.2 nM to 1 hr at 1 nM, with rupture becoming essentially immediate at >20 nM LEN (half-life of 2–3 min). Secondly, the survival curves are not purely exponential and show a lag period,

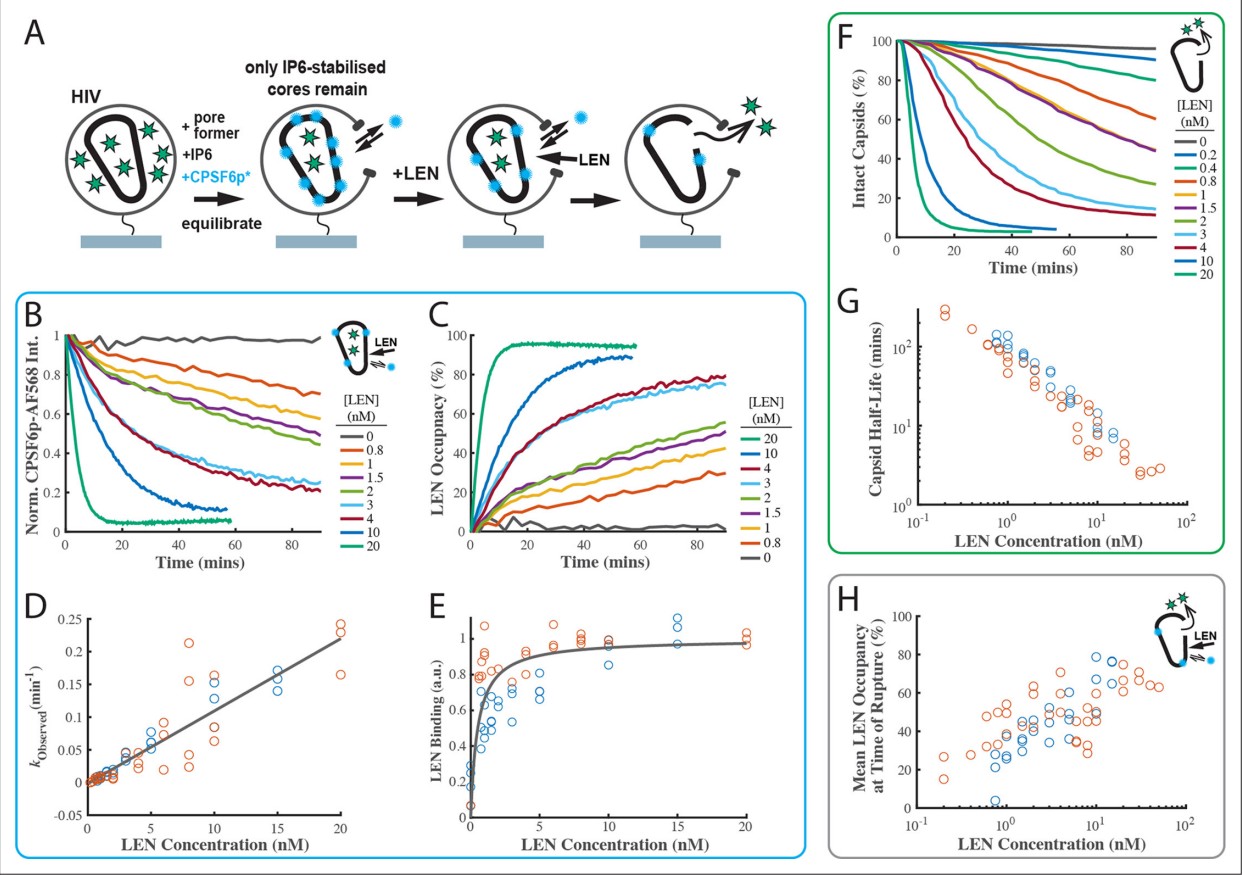

**Figure 4.** LEN binding to capsids is slow at concentrations close to the EC95 and limits the kinetics of LEN-induced capsid rupture. (**A**) Schematic diagram of the TIRF microscopy competition binding and capsid opening assay. GFP-loaded HIV attached to the glass coverslip are permeabilised using SLO and incubated in a solution 30 µM IP6. During this period, unstable capsids that do not respond to IP6 fall apart. Labelled CPSF6p (0.2 nM) functions as a paint probe (occupying ~2% of free FG binding pockets at equilibrium). LEN displaces labelled CPSF6p on the capsid over time, leading to a decrease of the labelled CPSF6p intensity. Rupture of the capsid is detected by GFP release. (**B–E**) Analysis of CPSF6p-AF568 signal disappearance to extract LEN binding kinetics and affinity. (**B**) Mean CPSF6p-AF568 displacement curves measured at all IP6-stabilised capsids in the field of views in the presence of 0–20 nM LEN. (**C**) LEN binding curves calculated from the CPSF6p-AF568 displacement curves in **B**. (**D**) Observed LEN binding rates (obtained from an exponential fit of the binding curves) as a function of LEN concentration. The slope of the linear fit to data from two experiments provides a LEN binding rate of $k_{on}$ = 1.84 ± 0.78×10$^5$ M$^{-1}$ s$^{-1}$. (**E**) Equilibrium binding curve for LEN on IP6-stabilised capsids. The fit (grey line) of an equilibrium binding model to the combined combined data from two experiments provided the dissociation constant of $K_D$ = 0.51 ± 0.58 nM. (**F–G**) Analysis of GFP release to determine the effect of LEN on capsid integrity. (**F**) Survival curves of IP6-stabilised capsids in the presence of 0–20 nM LEN. (**G**) Half life of the intact capsid as a function of LEN concentration determined from survival curves as the time point where half of the capsids had released GFP. (**H**) Combined analysis of the GFP release and labelled CPSF6p displacement. Mean LEN occupancy levels at the time of capsid rupture as a function of LEN concentration. Orange and blue circles in **D**, **E**, **G** and **H** correspond to two sets of experiments conducted with different viral preparations, different fluorophores, and different preincubation condition. Experiments shown as orange circles used CPSF6p-AF568, which was added to the capsid at the same time as LEN. Experiments shown as blue circles used CPSF6p-ATTO643, which was preincubated with HIV prior to LEN exposure. The CPSF6p displacement and capsid opening curves for the data set for the blue circles are shown in *Figure 4—figure supplement 3*.

The online version of this article includes the following figure supplement(s) for figure 4:

**Figure supplement 1.** The extent of the structural effect of LEN on the HIV capsid depends on LEN binding kinetics.

**Figure supplement 2.** Binding of labelled CPSF6p to closed capsids.

**Figure supplement 3.** Independent repeat of the single-molecule capsid uncoating experiment with fluorescent CPSF6p as a 'paint' probe to measure LEN binding.

especially at the lower end of the concentration range (discussed below). Thirdly, the overall efficiency of capsid rupture decreased sharply toward the lower end of the concentration range, dropping below 65% at 0.4 nM LEN (*Figure 4—figure supplement 3D*). We propose that the complex kinetics of LEN-induced capsid failure observed in our experiments are a convolution of (1) LEN concentration-dependent binding kinetics to gradually increase the occupancy of FG binding pockets with LEN

and (2) occupancy-dependent kinetics of capsid rupture where higher occupancy leads to the capsid becoming more unstable. It is also likely that capsids have heterogeneous susceptibility to LEN-induced structural damage, reflecting the heterogeneity of capsid architectures. This is supported by the decrease in breakage efficiency with LEN concentration (*Figure 4—figure supplement 3D*).

The power of using our combined CPSF6p paint and GFP release TIRF assay is that we can measure the LEN occupancy at the time of capsid opening and thus directly relate occupancy to drug-induced structural damage (*Figure 4H*). Interestingly, the mean LEN occupancy at the time of capsid rupture increased with LEN concentration. At sub-nM concentrations, LEN induced breakage when 20–30% of FG sites were drug-bound. At LEN concentrations ≥10 nM, the mean occupancy at the time of capsid breakage levelled off at ~70%. Taken together, our data suggest that LEN-induced capsid breakage can occur at low occupancies, albeit slowly. Breakage typically occurs at higher occupancies when LEN binding kinetics outpace the (occupancy-dependent) breakage kinetics. At high LEN occupancy (above 70%), capsids become so overwhelmingly unstable that it is practically impossible to bind LEN fast enough to get to complete occupancy before the capsid ruptures.

In summary, we conclude that efficient and rapid capsid breakage depends on reaching a sufficient occupancy of FG binding pockets, which itself depends on binding kinetics if drugs are added at

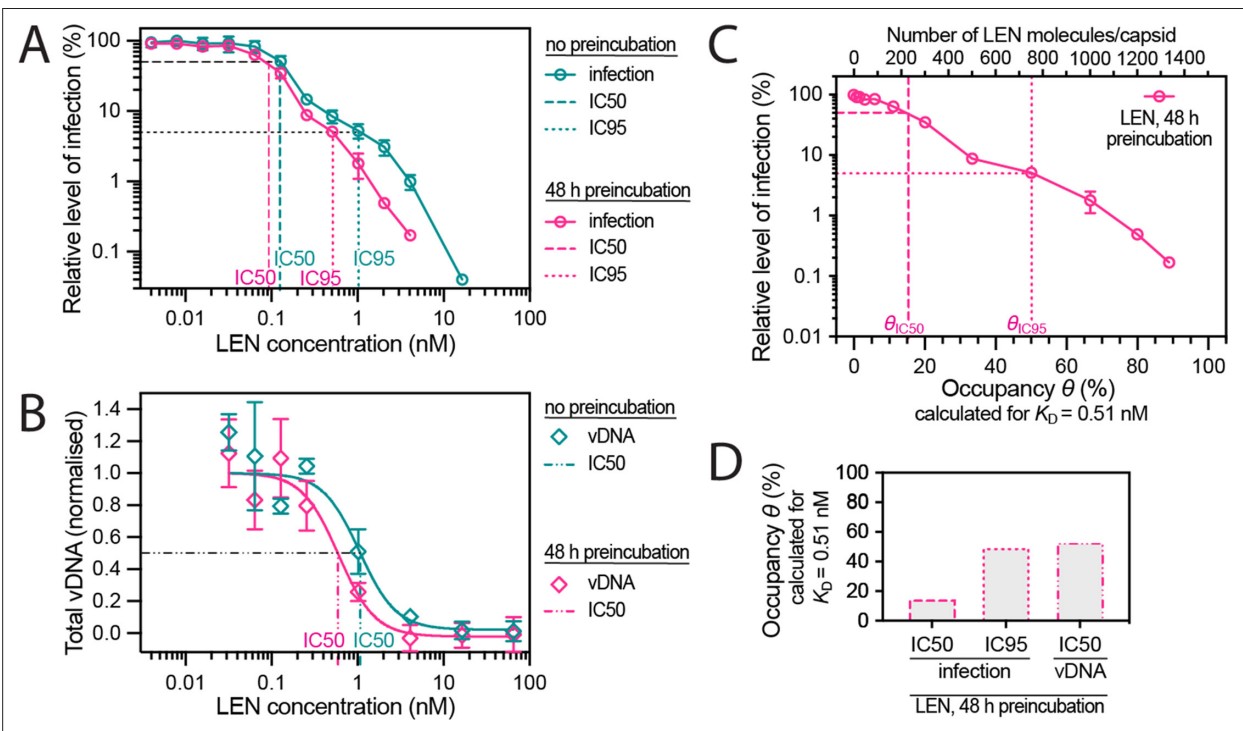

**Figure 5.** Preincubating HIV particles with LEN is required to obtain the full effect of the drug on capsid stability, HIV infection and reverse transcription. (**A**) Dose-response curves of Jurkat cells infected with VSV-G-pseudotyped GFP-encoding virus that was preincubated for 0 hr or 48 hr with the corresponding concentration of LEN. The number of infected cells was determined by flow cytometry 48 hr post infection. Data points represent percent infectivity relative to the vehicle control. The graph shows data from four (0 hr preincubation) or three (48 hr preincubation) independent experiments. The symbols represent the mean and the error bars the standard deviation. The vertical dashed lines indicate the IC50 values determined from least squares fits of the curves and the vertical dotted lines indicate the IC95 values determined as the concentration where the relative level of infection reaches 5% (0 hr preincubation: 1.024 nM [5.3% infection]; 48 hr preincubation: 0.512 nM [5.1% infection]). (**B**) Quantification using qPCR of total viral DNA from Jurkat cells collected 24 hr after infection with VSV-G-pseudotyped GFP-encoding virus as in panel D (with 0 hr or 48 hr preincubation with LEN). Normalised copy numbers (mean ± SD) from two independent experiments with two technical repeats each. Least squares fit (solid line) providing IC50 values of ~1080 pM without preincubation and ~590 pM with 48 hr preincubation. (**C**) Relative infection after 48 hr preincubation of HIV with varying LEN concentrations (same data as in A) plotted as a function of occupancy ($\theta$) of the FG binding pockets on the capsid. (**D**) Bar charts of occupancy at the IC50 and IC95 for infection and at the IC50 for vDNA synthesis. The occupancy in **C** and **D** was calculated for each LEN concentration assuming a dissociation constant ($K_D$) of 0.51 nM using the equation $\theta_{eq} = c_{LEN} / (c_{LEN} + K_D)$, where $\theta_{eq}$ is the occupancy at equilibrium and $c_{LEN}$ is the LEN concentration.

The online version of this article includes the following figure supplement(s) for figure 5:

**Figure supplement 1.** Preincubation of HIV with LEN.

the beginning of the experiment. At the lowest concentrations tested here (0.2 nM) LEN can break capsids, but slowly and only for a subset of capsids. In contrast, structural damage is induced on time scales that are relevant for infection in the range from high pM (>0.4 nM) to nM concentrations.

## Infection is inhibited at the stage of reverse transcription in the same concentration range that is required for efficient capsid rupture in vitro

Next, we tested whether preincubation of virions with LEN (to allow binding to the capsid to equilibrate prior to cell entry) would result in higher potency of the drug in Jurkat cells infected with VSV-G pseudotyped HIV encoding GFP as a reporter. Similar to the capsid-targeting drug PF74 (*Price et al., 2014*; *Saito et al., 2016*), the dose-response curves of LEN showed a biphasic inhibition profile, whereby the first phase (~0.1 nM) reduced infection to about 10% before a second inhibitory mechanism operating in the low nM range reduced infection levels to less than 1% (*Figure 5A*). Preincubation had little effect on inhibition during the first phase (1.3-fold reduction in IC50, *Figure 5—figure supplement 1A*), consistent with a block at a late stage (e.g. integration) such that the drug had sufficient time to reach the requisite occupancy irrespective of preincubation.

In contrast to the modest effect at low LEN concentrations, preincubation reduced infection by two- to threefold at concentrations ≥0.256 nM (i.e. during the second inhibitory phase), whereby the IC95 was 0.5 nM and 1 nM with and without preincubation, respectively (*Figure 5A*). The increased potency in this range after preincubation is consistent with a mechanism that depends on structural effects on the capsid, that are slow to manifest at sub-nM concentrations as shown above (*Figure 4H*).

Next, we determined the dose-dependent effect of LEN on reverse transcription using qPCR with primers amplifying total viral DNA (*Figure 5B*). The IC50 for reverse transcription in Jurkat cells was 1.08 nM LEN without preincubation, consistent with the dose-response in CEM cells (IC50=0.68 nM; *Sowd et al., 2021*). This value dropped to 0.59 nM when virions were preincubated for 48 hr with LEN prior to infection. Interestingly, the IC50 values for DNA synthesis corresponded to the IC95 values for infection, consistent with a second phase mechanism whereby loss of capsid integrity leads to inhibition of DNA synthesis.

Finally, the preincubation data allowed us to relate the early- and late-stage inhibition of infection and the inhibition of DNA synthesis to the occupancy of binding sites on the capsid since drug binding equilibrates during the preincubation period before the capsid is exposed to host cofactors. We used the $K_D$ value of the LEN-capsid interaction (0.51 nM, determined above by TIRF imaging) to replot the infection data as a function of occupancy (*Figure 5C*) and calculate the occupancy at IC50/IC95 for infection and at IC50 for DNA synthesis (*Figure 5D*). The corresponding plots calculated using the published $K_D$ for the LEN-CA hexamer interaction (0.24 nM) are shown in *Figure 5—figure supplement 1B and C* . From this analysis we conclude that LEN inhibits a late post-entry step when binding to 15% ($K_D$ = 0.51 nM) or 30% ($K_D$ = 0.24 nM) of sites, but this block is insufficient to reduce infection below a level of ~10%. At 50% ($K_D$ = 0.51 nM) or 70% ($K_D$ = 0.24 nM) occupancy of sites on the CA lattice, LEN additionally blocks infection by inhibiting reverse transcription, ultimately reducing infection to levels below 1%.

## LEN and IP6 synergise to promote CA assembly but compete to bias the assembly pathway towards tube (LEN) versus cone (IP6) formation

Since both IP6 (*Dick et al., 2018*; *Renner et al., 2021*) and LEN (*Bester et al., 2020*; *Link et al., 2020*) promote CA assembly in vitro, we tested how the combination of these molecules affects CA assembly in low salt conditions. We monitored the assembly kinetics of recombinant CA by measuring the absorbance at 350 nm and collected samples at the end of each experiment for analysis by negative stain electron microscopy (*Figure 6*). First, we varied the concentration of IP6 and observed that CA (75 μM) assembled with similar kinetics in the presence of 100 μM or 150 μM IP6 but did not assemble at lower IP6 concentrations (*Figure 6A*, left). As expected, negative stain EM images of assembly products formed with only IP6 showed primarily conical shapes with dimensions similar to those observed for native HIV capsids (*Figure 6A*, right). When we repeated this titration in the presence of 50 μM LEN (substochiometric relative to CA), we observed CA assembly across the entire concentration range (10–150 μM IP6), with IP6 accelerating assembly kinetics and yields in a concentration-dependent manner (*Figure 6B*, left). Strikingly, LEN in the presence of low IP6 promoted formation of CA tubes (often closed at their ends) with lengths of >500 nm, while

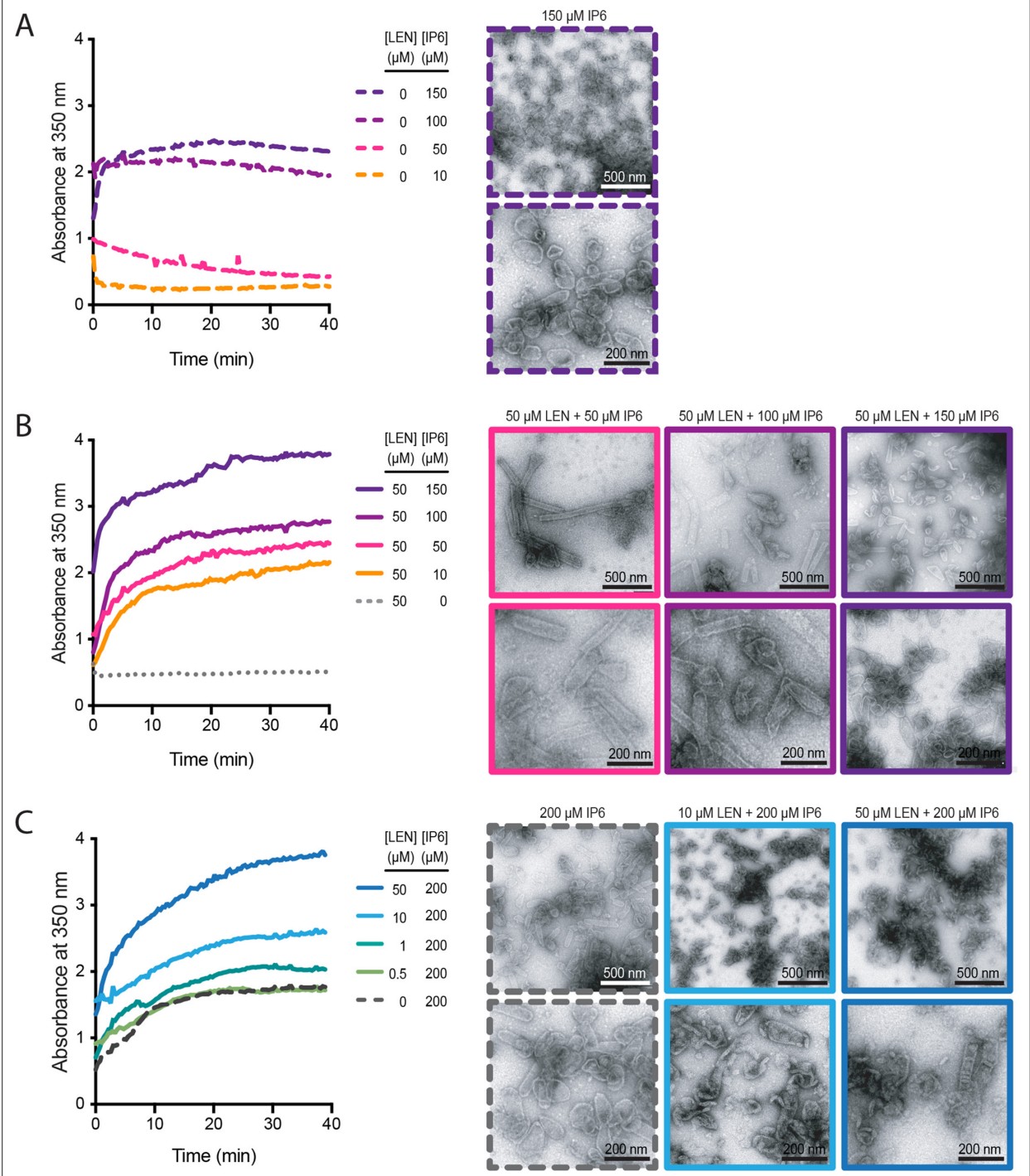

**Figure 6.** IP6 and LEN synergise to promote CA assembly. In vitro assembly reactions of CA (75 μM) were carried out in 50 mM MES (pH 6.0) containing 1 mM DTT and monitored in real time by absorbance measurements at 350 nm. The assembly products obtained at the end of the reaction were imaged by negative staining electron microscopy. (**A, B**) Assembly kinetics (left) and products (right) formed at 10–150 μM IP6 in the absence (**A**) or the presence (**B**) of 50 μM LEN. (**C**) Assembly kinetics (left) and products (right) formed at 0–50 μM LEN in the presence of 200 μM IP6.

The online version of this article includes the following figure supplement(s) for figure 6:

**Figure supplement 1.** LEN promotes assembly of additional CA structures outside the capsid in mature HIV particles.

increasing IP6 concentrations biased assembly increasingly towards shorter tubes and conical shapes (*Figure 6B*, right). Next, we varied the concentration of LEN in the presence of 200 µM IP6. Addition of 1–50 µM LEN increased CA assembly efficiency in a concentration-dependent manner above the level observed for IP6 only (*Figure 6C*, left). Notably, the highest LEN concentration (50 µM) did not promote CA (75 µM) assembly without IP6 in low salt conditions (*Figure 6B*, left). Negative stain EM images confirmed cone formation in the presence 200 µM IP6 and showed that addition of drug led to the formation of aberrant and broken structures in a concentration-dependent manner (*Figure 6C*, right). We conclude that LEN is insufficient to promote CA assembly by itself in low salt conditions but synergises with IP6 to increase assembly kinetics and yields. Importantly, IP6 promotes cone assembly, whereas LEN biases assembly toward tube formation such that closed tubes form in the presence of high drug and sufficiently low IP6. When both molecules are present at high concentrations, where neither molecule can dominate the assembly pathway, assembly proceeds in an aberrant fashion yielding heterogeneous structures. Taken together these observations show that IP6 and LEN synergise to promote assembly but preferentially stabilise different CA lattice structures.

## LEN promotes CA overassembly inside mature virions

As the single-molecule TIRF analysis suggested that LEN alters capsid properties inside the intact virion, we used cryo-electron tomography (cryoET) and carried out 3-dimensional reconstructions on untreated and drug-treated (700 nM) virions, which showed the expected distribution of maturation states and capsid shapes (*Figure 6—figure supplement 1A*) as observed before (*Fontana et al., 2016*; *Mallery et al., 2021*; *Mattei et al., 2014*; *Renner et al., 2021*). The tomograms of LEN-treated virions did not reveal obvious capsid defects (such as large holes), and we were not able to assign small apparent discontinuities as defined holes in the capsid at the resolution of our tomograms. In contrast to untreated virions which contained mostly single capsids comprised of a single CA layer (74%), almost all LEN-treated virions (96%) contained additional CA structures next to the main capsid (61%) and/or appeared with a double layered capsid (45%) (*Figure 6—figure supplement 1B*). These observations suggest that LEN induces assembly of the pool of free CA that is otherwise not incorporated into the capsid. CypA paint analysis of virions pre-treated with 500 nM LEN corroborated this overassembly phenotype, showing an average 1.7–1.8-fold increase of the CypA paint signal (*Figure 6—figure supplement 1C–E*), consistent with the presence of a larger overall CA lattice surface area. Since IP6 is enriched in HIV particles (reaching concentrations of ~500 µM; *Mallery et al., 2018*), the drug-cofactor synergy driving aberrant assembly inside virions recapitulates the in vitro assembly data described above. This overassembly defect is likely to also play out during capsid assembly in virions that undergo maturation in the presence of drug, consistent with the observation that virions produced in the presence of LEN (*Link et al., 2020*) or the closely related compound GS-CA1 (*Yant et al., 2019*) contain improperly shaped capsids.

## PF74 but not BI-2 slows CA lattice disassembly after capsid rupture

We have shown before using single-molecule TIRF uncoating assays that PF74, an HIV inhibitor that binds to the same site as LEN, strongly accelerates capsid opening and stabilises the lattice of the capsid thereafter (*Márquez et al., 2018*). The concentration (10 µM) used in those experiments is 40–80-fold above the $K_D$ determined for the interaction with CA hexamers (between 0.12 µM *Price et al., 2014* to 0.26 µM *Bhattacharya et al., 2014*), such that 97–99% of binding sites of the capsid are predicted to be occupied with a drug molecule. PF74 is an important tool to study HIV capsid-associated processes but has been described to either promote capsid uncoating (*Da Silva Santos et al., 2016*; *Selyutina et al., 2022*; *Shi et al., 2011*) or to stabilise capsids (*Rankovic et al., 2018*) or to have no effect on capsid integrity (*Hulme et al., 2015*). To resolve this ambiguity and further characterise the ability of PF74 to stabilise (broken) CA lattices, we carried out CypA paint experiments in the presence of 0.1–10 µM PF74 (*Figure 7A*). In this concentration range, binding quickly reaches equilibrium and occurs before membrane permeabilisation in our assay. PF74 slowed the loss of CA from leaky and opening capsids in a concentration-dependent manner. Maximum stabilisation required essentially all sites to be drug-bound (97–99% at 10 µM), but even at this occupancy we observed ~70% signal loss after 80 min (*Figure 7B*). Thus, PF74 was unable to prevent CA lattice disassembly to the same extent as LEN (*Figure 3E*). PF74 also required near saturating occupancy to

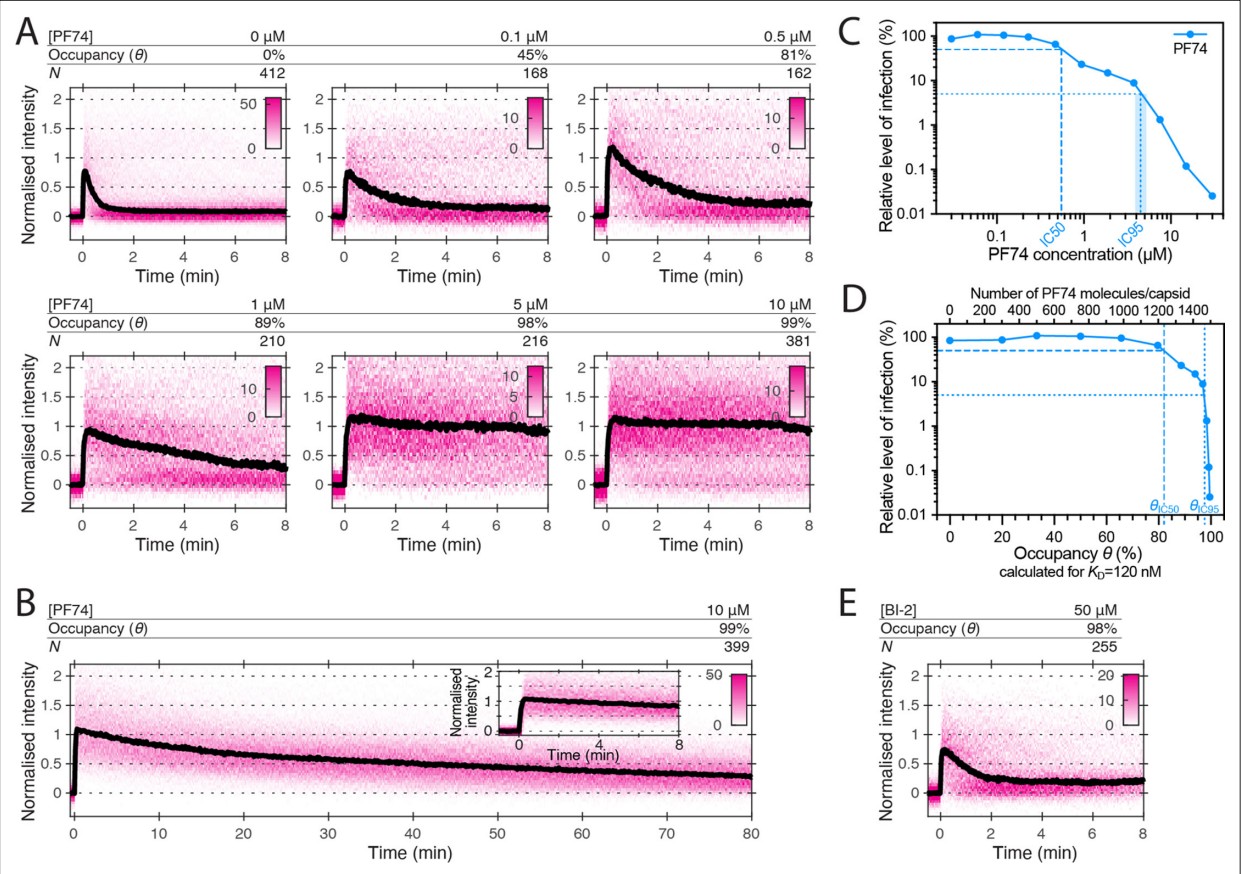

**Figure 7.** Effects of PF74 and BI-2 on CA lattice stability. (**A, B**) Heatmaps (magenta) and median traces (black line) of the CypA intensity measured at particles with leaky or opening capsids in the presence of 0–10 μM PF74 showing that PF74 stabilises the CA lattice in a concentration-dependent manner. (**A**) PF74 titration in 8 min experiments. (**B**) 10 μM PF74 in an 80 min experiment showing that capsid lattices disassemble over this time period. The inset shows the first 8 min of the trace. (**C**) Dose-response curve of Jurkat cells infected with VSV-G-pseudotyped GFP-encoding virus in the presence of the indicated concentrations of PF74. The number of infected cells was determined by flow cytometry 48 hr post infection. Data points represent percent infectivity relative to the vehicle control. The dashed line indicates the IC50 determined from least squares fits of the curve (~0.6 μM, corresponding to ~5×$K_D$). The dotted line indicates the IC95 (4–5 μM, corresponding to 30–40×$K_D$). (**D**) Relative infection (same data as in **C**) plotted as a function of occupancy ($\theta$) of the FG binding pockets on the capsid calculated for each PF74 concentration assuming a $K_D$ of 120 nM. (**E**) Heatmap (magenta) and median trace (black line) of the CypA intensity at particles with leaky or opening capsids in the presence of 50 μM BI-2 showing that BI-2 does not prevent CA lattice disassembly.

The online version of this article includes the following figure supplement(s) for figure 7:

**Figure supplement 1.** PF74 and BI-2 binding at high occupancy leads to capsid rupture.

accelerate capsid opening, and at a concentration of 1 μM (79–89% occupancy), the drug showed only partial capsid-breaking activity (***Figure 7—figure supplement 1***).

The FG pocket-binding drug BI-2 also has potent capsid-breaking activity (***Márquez et al., 2018***) when used at a concentration (50 μM) that is ~40-fold above the $K_D$ (1.2 μM) of the interaction with CA hexamers (***Price et al., 2014***). We predicted that BI-2 would be unable to stabilise CA lattices after capsid opening since the compound contacts only one of the two CA subunits forming the FG binding pocket (***Figure 1A***). As expected, CypA paint analysis showed that 50 μM BI-2 only marginally slowed the release of CA subunits from the lattice (***Figure 7E***). These observations together with those described above suggest that PF74 exerts the same structural effects on the capsid as LEN but less potently, while BI-2 breaks the capsid but is unable to slow subsequent disassembly.

To relate the degree of the capsid-altering effects of PF74 to its effects on HIV infection, we measured the dose-response curve for inhibiting infection of Jurkat cells with VSV-G pseudotyped HIV encoding GFP as a reporter (***Figure 7C***). The curve showed the characteristic biphasic profile with the first phase levelling off at ~10% infection at ~1 μM followed by a second drop in infection

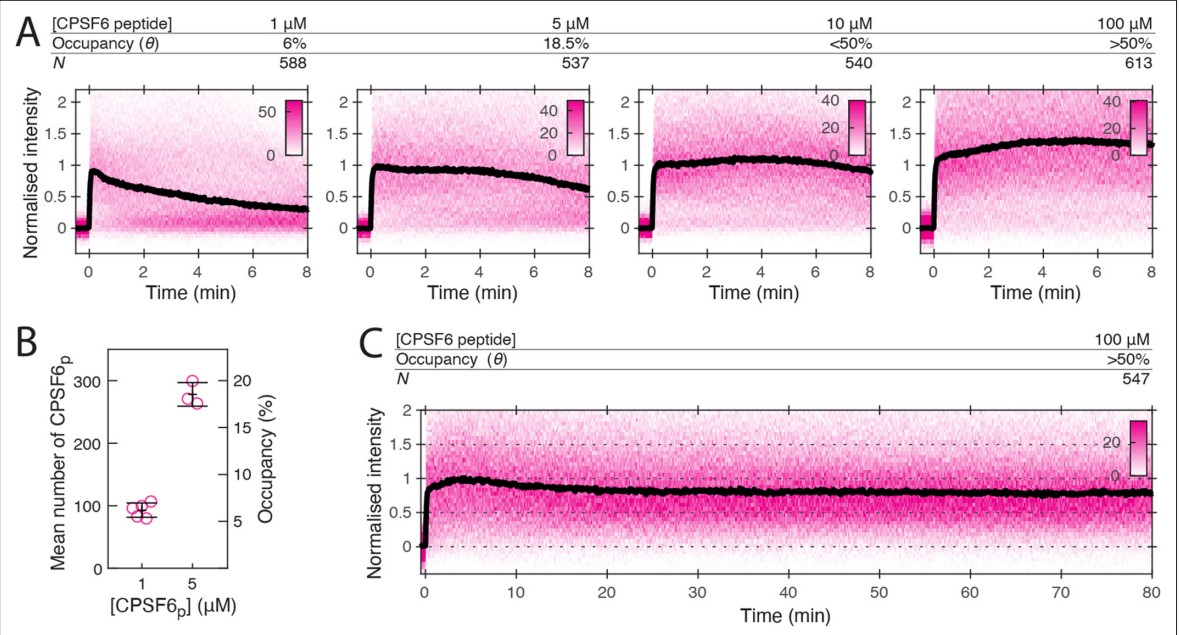

**Figure 8.** CPSF6 peptide stabilises the CA lattice at low occupancy. (**A**) Heatmaps (magenta) and median traces (black line) of the CypA intensity measured at particles with leaky capsids in the presence of 1–100 µM CPSF6p showing that the peptide stabilises the CA lattice at concentrations below the $K_D$ of the peptide-CA hexamer interaction. The occupancy ($\theta$) at 1 µM and 5 µM was determined experimentally (see **B**) while $\theta$ was estimated for higher concentrations assuming a $K_D$ of 50 µM. (**B**) Mean number of CPSF6p bound per capsid determined from CPSF6p-AF568 binding experiments. Each symbol represents an independent binding experiment, black bars indicate mean and standard deviation. (**C**) Heatmap (magenta) and median trace (black line) of the CypA intensity at particles with leaky or opening capsids in the presence of 100 µM CPSF6p monitored over 80 min, showing long-term stabilisation of the CA lattice.

The online version of this article includes the following figure supplement(s) for figure 8:

**Figure supplement 1.** Effect of CPSF6p on capsid opening.

above 4 µM. To facilitate comparison to the corresponding biphasic LEN data, we replotted the infection data as a function of occupancy (*Figure 7D*). Compared to LEN, the PF74 curve was shifted to higher occupancies, whereby late phase inhibition with PF74 required >95% occupancy, i.e. the same as required for stabilisation of open CA lattices (*Figure 7A*). In comparison, LEN required between 35–70% occupancy for late phase inhibition (*Figure 5C* and *Figure 5—figure supplement 1B*), again corresponding to the level required for CA lattice stabilisation (*Figure 3E*). Taken together, our CypA paint and infection data show that LEN can elicit the detrimental capsid structural changes characteristic of the second phase at lower occupancy than PF74.

## CPSF6 peptide stabilises the CA lattice at low occupancy

Given that different compounds can influence uncoating mechanisms in different ways, it is conceivable that the virus uses endogenous ligands to tune capsid stability. We therefore sought to compare the effect of the above drugs with CPSF6, a host cell protein which also binds the FG-pocket, and bridges the gap between monomers (*Price et al., 2012*). CPSF6 colocalises with the capsids at the nuclear pore complex and inside the nucleus where it plays a role in facilitating nuclear entry and dictating integration site position (*Bejarano et al., 2019*; *Schaller et al., 2011*; *Sowd et al., 2016*; *Zila et al., 2021*). To investigate the effect of CPSF6 on capsid stability, we performed single-molecule TIRF uncoating experiments in the presence of the minimal CPSF6 peptide (CPSF6p) that binds to the FG pocket (*Figure 1D*). At 100 µM (twofold above the KD of 50 µM for binding to CA hexamers), CPSF6p promoted capsid opening, but with slower kinetics than the FG pocket-binding drugs (*Figure 8—figure supplement 1*). CypA paint analysis showed that 100 µM CPSF6p strongly inhibited CA release (*Figure 8A*) as evident from slow decay in the CypA paint signal over 80 min (*Figure 8C*). Further, 1 µM CPSF6p (6% occupancy, *Figure 8B*) was insufficient to stabilise the lattice but 5 µM (18.5% occupancy, *Figure 8B*) partially and 10 µM (<50% occupancy) strongly inhibited CA lattice dissociation.

Thus, CPSF6p was similarly stabilising relative to occupancy to LEN and more potent than PF74 and BI-2, providing partial stabilisation when fewer than 20% of binding sites are occupied.

## Discussion

Here, we show that the antiretroviral LEN has two opposing effects on the HIV-1 capsid at high occupancy: it prevents dissociation of CA from the lattice but induces capsid rupture. This apparently counterintuitive phenomenon is consistent with a model in which the lattice must be simultaneously stable but flexible. The antiviral mechanism of LEN can therefore be considered a form of 'lethal hyperstability' in which lattice stability is increased at the cost of its flexibility and ultimately capsid integrity. This is reminiscent of the 'lethal mutagenesis' mechanism of antiviral polymerase drugs such as favipiravir (*Perales et al., 2011*; *Arias et al., 2014*). Just as with capsid stability, viral replication has conflicting requirements – errors are necessary to promote mutagenesis and evolvability (*Tokuriki and Tawfik, 2009*) but too many errors and fidelity is compromised. Lethal mutagenesis drugs exert their antiviral affects by pushing viral polymerases to these unsustainable error rates. Similarly, lethal hyperstability capsid drugs like LEN push the capsid lattice to extremely stabilised structures that are incompatible with capsid integrity. Importantly, this mechanism dominates at drug doses required to suppress replication to clinically relevant levels (*Dvory-Sobol et al., 2022*).

The effects of lethal hyperstability are observable prior to reaching binding equilibrium or full occupancy and we estimate that efficient and rapid drug-induced capsid rupture and lattice stabilisation occurs when half of the binding sites are occupied. The overall kinetics of inducing structural effects depend on LEN binding kinetics and occupancy-dependent structural changes. As occupancy increases over time, breakage becomes increasingly likely and rapid. Due to the high affinity of the drug, sub-nM concentrations are sufficient to induce structural effects, but it takes hours to reach the requisite binding level. The cofactor IP6, which, in the absence of drug, would normally delay spontaneous capsid opening by many hours, counteracts but is ultimately unable to prevent LEN-induced capsid rupture. It is also worth noting that HIV capsids are pleiomorphic and exhibit different levels of intrinsic stability as evident from our single-molecule analysis, such that the occupancy required for LEN-induced structural changes is likely to vary between different capsid architectures.

The comparison of the single-molecule analysis of the three major drugs (BI-2, PF74, and LEN) offers insight into how binding stabilises the CA lattice, even when it is no longer a closed shell. BI-2 is the simplest compound, and only makes contacts within a single monomer and has the lowest affinity. As such it offers no potential for enhancing contacts between individual proteins within the lattice. It is therefore unsurprising that, even at 50 μM (approaching the solubility limit of the drug), BI-2 provides no significant lattice stabilisation once the capsid has ruptured. PF74 and LEN, on the other hand, both make contacts across the junction between monomers within the context of the hexamer, and both have been shown to have higher affinity for hexamers than monomers. This suggests that the two drugs lock monomers together within the hexamer, potentially limiting their ability to move with respect to each other - reducing degrees of freedom and hence flexibility within the lattice that is required to maintain a closed capsid. Evidence that the hexamers are stabilised comes from lattice assembly experiments in which LEN promotes CA assembly in the presence of IP6 (*Figure 6*) or high salt (*Bester et al., 2020*; *Link et al., 2020*). While PF74 does stabilise the CA lattice post-rupture, the lattice will deteriorate on a time scale of an hour. LEN-treated cores, on the other hand, show no lattice loss even after 5 hr post-rupture. The degree of lattice stabilisation, therefore, correlates with the degree to which the drugs bridge the junction between monomers.

How exactly stabilising the lattice triggers capsid rupture is less clear. It has previously been suggested that the binding of the drug reduces the flexibility of the CA molecule and therefore the lattice plasticity required to maintain capsid integrity (*Bhattacharya et al., 2014*). These authors also suggest that, due to its hexamer preference, PF74 stabilises hexamers at the cost of pentamers. As precisely 12 pentamers are required, any process that disfavour pentamers, would necessarily lead to capsid rupture. A recent cryoEM study comparing the structures of hexamers and pentamers identified residues 58–61 as being a potential 'switch' between the two states (*Schirra et al., 2023*); this switch was also confirmed in a second cryoEM study (*Stacey et al., 2023*). Importantly, the pentamer configuration of these residues results in a remodelling of the FG-binding site. The authors speculate that LEN may result in an induced fit switch to the hexameric configuration (*Schirra et al., 2023*), and our data would be consistent with such a model. Another cryoET study of HIV cores incubated under

conditions that facilitate reverse transcription inside the capsid showed that treatment with GS-CA1 (an analogue of LEN) for 4 hr led to loss of CA lattice pieces and a flattening of the remaining lattice (*Christensen et al., 2020*). The authors proposed that the compound restricts CA flexibility, causing a build-up of lattice strain, and consequently lattice fracture. We previously proposed a similar mode of action for the capsid-breaking activity of PF74 (*Márquez et al., 2018*). Finally, cryoEM of templated CA lattices confirmed that LEN binds exclusively to hexamers and suggested a critical role of IP6 for the formation of pentamers and high curvature (*Highland et al., 2023*), consistent with our observation that IP6 can partially antagonise LEN-induced rupture of the closed cone.

LEN also possesses antiviral activity during viral production. Not only does LEN compromise the integrity of existing capsid cores but it interferes with assembly/maturation as well. Our in vitro assembly experiments suggest that, while LEN promotes IP6-driven assembly it leads to improperly assembled cones that cannot be closed. This is consistent with the observation that LEN (*Bester et al., 2020*) and the closely related analogue GS-CA1 (*Yant et al., 2019*) lead to the formation of aberrant capsids in virions. IP6 binds to the electropositive 'pore' created by 6 (or 5) copies of Arg18 at the centre of each CA hexamer (or pentamer). By neutralising the charge repulsion, IP6 stabilises these structures and is thought to be particularly important for incorporation of pentamers (*Gupta et al., 2022*; *Renner et al., 2021*) required for forming the high curvature lattice at the ends of the cone. On the other hand, LEN drives assembly of low curvature hexameric lattices (*Figure 6B*). Both compounds together might lead to uncontrolled lattice growth without the ability to reverse defects and/or LEN might drive remodelling of pentamers at the growing lattice edge into hexamers (*Grime et al., 2016*). Furthermore, the over-assembly phenotype that we observe upon drug treatment of mature virions supports the notion that LEN causes aberrant CA assembly in addition to distorting existing structures. While our data do not resolve how the LEN-induced capsid defects manifest structurally in isolated capsids, within mature virions this could be due to a build-up of strain within the primary capsid. In addition, 'secondary CA lattices' would likely compete for the limited IP6 present within the virion, effectively reducing the amount available to the primary capsid, thereby reducing its resistance to rupture and possibly also adjusting the hexamer/pentamer balance.

The antiviral multimodality of LEN is also seen in the biphasic response curve in the infection assay. The high-dose phase occurs above 0.5 nM LEN, above which the considerations of capsid rupture are relevant. The high-dose phase also corresponds to the concentrations at which reverse transcription is also inhibited. A biphasic inhibition curve has previously been seen for PF74 (and is repeated here) where, similarly, loss of reverse transcription accompanies this high-dose phase. We and others had previously attributed the loss of reverse transcriptase activity to the opening of the capsid lattice and release of the RT enzyme (*Christensen et al., 2020*; *Jennings et al., 2020*; *Sowd et al., 2021*). Our data support this same explanation for LEN-induced loss of RT. However, LEN is not simply a tighter-binding version of PF74. *Figure 5C* (LEN) and *Figure 7D* (PF74) show that when the drug response curves are plotted as a function of occupancy, LEN exerts its effects at much lower capsid occupancy than PF74. In a previous study, when BI-2 and PF74 were similarly compared, they were found to be identical (*Price et al., 2014*). This suggests that increased CA affinity only partially explains LEN's superior potency and may speak to a greater 'rigidification potential' or ultrastructure-altering capability relative to prior compounds that target this same site.

While the second phase of the dose response curve has a clear explanation, the activity of the drugs at the first phase remains controversial. At these concentrations (<0.5 nM for LEN;<4 nM PF74,<50 nM for BI-2) neither capsid rupture nor loss of viral DNA synthesis are observed. One possible explanation could be that at low drug concentration the low occupancy contributes to CA lattice stability, without the concomitant rupture observed at higher doses. In combination with the cone-stabilising activity of IP6, low dose LEN could render the capsid core too stable thereby leading to altered genome release kinetics during infection. However, the observation that 2-LTR circle formation is not reduced at the EC50 for infection (*Bester et al., 2020*) is inconsistent with this model as failure to release the viral cDNA from the capsid would be expected as a result of capsid stabilisation. Alternatively, disrupting the ability of the capsid to interact with FG-containing cofactors (Sec24C, Nup153, and CPSF6) has also been proposed to inhibit infection. Our data predict that, at its EC50, LEN occupies approximately 15-30% of the FG-binding sites, while PF74 occupies >80% of sites (*Figure 7D*). However, the true degree to which these drugs are able to compete with cofactor binding during infection is complicated by the spatial organisation of the cofactors in the cell, the unknown degree to which they

compete with each other, the possibility that they exist as high-avidity multimers, and fluctuations in their abundances throughout the cell cycle and between cell types (*Wei et al., 2022*).

The fact that these drugs target a cofactor binding site (the FG pocket) and are capable of modulating capsid rupture and stability raises the question as to whether the virus is employing these cofactors to regulate uncoating. This has previously been suggested for cofactors binding at other CA lattice sites, such as the interactions between CypA and the cyclophilin loop (*Rasaiyaah et al., 2013*) or between IP6 and the R18 ring (*Mallery et al., 2018*). The best characterised of the cofactors interacting with the FG pocket is CPSF6, which binds via a linear peptide motif and, like PF74 and LEN, makes contacts across the junction between monomers. Our observation that CPSF6p stabilises the CA lattice is consistent with the notion that this bridging interaction promotes stability. A surprising result was the degree to which CPSF6p provided stabilisation, achieving this effect when as few as 20% binding sites are occupied. This could indicate that the nature of the bridging interaction is somehow more flexible than those observed for the drugs, leading to differences in the dependence of binding on the curvature of the CA lattice. Indeed, cryoET imaging of CA cones have shown that CPSF6 binding is independent of lattice curvature (*Stacey et al., 2023*) while PF74 prefers low curvature regions (*Lu, 2020*). We acknowledge that we are studying an isolated monomeric peptide motif, while endogenous CPSF6 will likely have a higher binding constant due to avidity, as it is expected to be at least a dimer (*Ning et al., 2018*), but possibly higher, as CPSF6 has been associated with phase separated condensates within the nucleus (*Greig et al., 2020*). The nuclear localisation of CPSF6 and the above stabilisation activity may account for the observation of capsid remnants in the nucleus discrete from integrated proviral DNA (*Müller et al., 2021*; *Zila et al., 2021*).

For decades, the concept that the HIV capsid must release its contents to complete infection has been accepted, and the metastability of the capsid has been recognised as critical for the viral life cycle. LEN is the first capsid-targeting drug for treatment of HIV infection. Our work here shows that LEN functions by lethal hyperstabilisation, and that this is a powerful mechanism for achieving multi-log impacts on viral infectivity. As such, this mechanism and the tools that we and others have developed for studying it will likely be relevant to the development of new therapeutics targeting a range of viral infections. Furthermore, LEN reveals that FG-pocket binding can drastically alter the capsid integrity. The exact nature of the induced ultrastructural defect warrants further study, as does the role of single and multiple endogenous cofactors on the uncoating process.

## Materials and methods

**Key resources table**

| Reagent type (species) or resource | Designation | Source or reference | Identifiers | Additional information |
|---|---|---|---|---|
| Cell line (human) | HEK293T | ATCC | ATTC:CRL-3216 | |
| Cell line (human) | Jurkat, Clone E6-1 | ATCC | ATTC:TIB-152 | |
| Recombinant DNA reagent | pNL4.3-iGFP-ΔEnv | DOI: 10.1371/journal.ppat.1002762 | | |
| Recombinant DNA reagent | psPAX2 | NIH AIDS Reagent Program | NIH AIDS Reagent Program:#11348 | |
| Recombinant DNA reagent | pCRV1-GagPol | DOI: 10.1128/jvi.78.21.12058–12061.2004 | | |
| Recombinant DNA reagent | pCSGW | DOI: 10.1126/science.272.5259.263 | | Transfer vector encoding GFP |
| Recombinant DNA reagent | pMD2.G | Addgene (Trono lab) | Addgene:#12259 | Encodes VSV-G envelope protein |
| Recombinant DNA reagent | pMCSG7-DLY | DOI: 10.1128/iai.00927–12 | | Plasmid for bacterial expression of recombinant desulfolysin from Desulfobulbus propionicus |

*Continued on next page*

*Continued*

| Reagent type (species) or resource | Designation | Source or reference | Identifiers | Additional information |
|---|---|---|---|---|
| Sequence-based reagent | 2-LTR junction fwd | Integrated DNA Technologies | | PCR primer for quantification of 2-LTR circles, 5'-GCTAACTAGGGAACCCACTGCTTAAG-3' |
| Sequence-based reagent | 2-LTR junction rev | Integrated DNA Technologies | | PCR primer for quantification of 2-LTR circles, 5'-ACTGGTACTAGCTTGTAGCACCATCCA-3' |
| Sequence-based reagent | Mf374 probe | Sigma-Aldrich | | 6-FAM-ACA [C]A[C]A[A]G[G][C]T-BHQ-1 |
| Sequence-based reagent | KBrun692F | Sigma-Aldrich | | PCR primer for quantification of viral DNA, 5'-CAGGACTCGGCTTGCTGAAG-3' |
| Sequence-based reagent | KBrun797R | Sigma-Aldrich | | PCR primer for quantification of viral DNA, 5'-GCACCCATCTCTCTCCTTCTAGC-3' |
| Peptide, recombinant protein | CPSF6p | GenScript | | custom synthesised peptide, sequence: PVLFPGQPFGQPPLG |
| Peptide,-recombinant protein | CPSF6p-Cys | GenScript | | custom synthesised peptide, sequence: PVLFPGQPFGQPPLGC |
| Chemical compound, drug | GS-6207 | MedChemExpress | MedChemExpress:HY-111964 | |
| Chemical compound, drug | PF74 | Sigma-Aldrich | Sigma-Aldrich:SML0835 | |
| Chemical compound, drug | BI-2 | Enamine | Enamine:Z1982491200 | |
| Chemical compound, drug | EZ-Link Sulfo-NHS-LC-LC-Biotin | Thermo Fisher Scientific | Thermo Fisher Scientific:21338 | |
| Software, algorithm | JIM Immobilized Microscopy analysis package | *Walsh, 2021* | | https://github.com/lilbutsa/JIM-Immobilized-Microscopy-Suite |
| Other | SLO | Sigma-Aldrich | Sigma-Aldrich:S5265-25KU | Streptolysin O from *Streptococcus pyogenes* |

## Cell lines

HEK-293T cells and Jurkat cells were obtained from ATCC. Identity testing was carried out by PCR. Cell lines tested negative for mycoplasma.

## Production of GFP-loaded HIV particles for TIRF uncoating experiments

HIV particles lacking envelope protein were produced, biotinylated and purified as described (*Márquez et al., 2019*). Briefly, HEK-293T cells were transfected using PEI with a mixture of the plasmids pNL4.3-iGFP-ΔEnv and psPAX2 (1.4:1, mol/mol) to produce GFP-loaded HIV particles or with a mixture of pCRV1-GagPol and pCSGW (1:1.7, mol/mol) to produce dark HIV particles. The medium was exchanged 18 hr post transfection and the virus-containing medium was collected 72 hr post transfection and centrifuged (2100 x *g*, 20 min, 4 °C) to remove cells. The viral particles were then biotinylated using EZ-Link Sulfo-NHS-LC-LC-Biotin and purified by size exclusion chromatography.

## Expression and purification of DLY

The gene encoding the pore-forming protein desulfolysin (DLY) was subcloned by ligation independent cloning into the pMCSG7 vector from the pET22b construct described in *Hotze et al., 2013* to introduce an N-terminal His-tag. DLY was expressed in *E. coli* BL21(DE3) pREP4 cells in TB media with Ampicillin (100 mg/mL) by induction with 0.2 mM IPTG at 37 °C with shaking for 4 hours. Cells were harvested by centrifugation and lysed in 20 mM Tris pH 7.2 300 mM NaCl buffer with 10% glycerol,

protease inhibitor, 0.1% Triton-X100, DNAse and lysozyme for 1 hr at room temperature. The lysate was clarified by centrifugation and passed over a HisTrap HP column (Cytiva Life Sciences) equilibrated in 20 mM Tris pH 7.2, 300 mM NaCl and 5% glycerol. DLY was eluded over a linear gradient of 0–500 mM imidazole. The eluted protein was further purified by size exclusion chromatography on a HiLoad 16/60 Superdex 200 pg column (Cytiva Life Sciences) preequilibrated with 20 mM Tris pH 7.2, 300 mM NaCl, 5% glycerol, 0.5 mM DTT.

## Expression and purification of CypA

Human CypA was expressed in BL21(DE3) *E. coli* for 3 hr after IPTG induction in LB medium at 37 °C with shaking. Cells were harvested by centrifugation and lysed by sonication on ice in a buffer containing 25 mM HEPES, pH 7.6, 1 mM DTT, 0.02% $NaN_3$, 'Complete' protease inhibitor and 1 mg $mL^{-1}$ lysozyme. The lysate was clarified by centrifugation. CypA was purified by subtractive anion exchange chromatography using a 10 mL HiTrap Q HP column (GE Healthcare Life Science) equilibrated with 25 mM HEPES, pH 7.6, 1 mM DTT, 0.02% $NaN_3$. CypA fractions eluting in the flow-through were adjusted to pH 5.8 with 1% v/v acetic acid, centrifuged and applied to a cation exchange chromatography column (5 mL HiTrap SP HP, GE Healthcare Life Science) equilibrated with 25 mM sodium phosphate, pH 5.8, 1 mM DTT, 0.02% $NaN_3$. CypA was eluted with a linear gradient from 0 to 1 M NaCl over 20 column volumes. CypA was dialyzed against storage buffer (25 mM MOPS, pH 6.6, 1 mM DTT, 0.02% $NaN_3$), concentrated using an Amicon-15 Ultra centrifugal filtration device (10 k MWCO, Merck) and frozen in liquid nitrogen for storage at –80 °C.

## Labelling of CypA

CypA was dialysed against PBS (pH 7.4, 0.1 mM TCEP) and labelled by reaction with a fourfold molar excess of Alexa-Fluor 568-C5-maleimide (Thermo Fisher Scientific, A10254) for 10 min at room temperature. The reaction was quenched by addition of DTT. Labelled CypA was separated from unconjugated dye using Zeba desalting spin columns (Thermo Fisher Scientific) equilibrated with 50 mM Tris, pH 7.9, 20% v/v glycerol, 1 mM DTT. Under these conditions, CypA is quantitatively labelled at residue C51. Labelled CypA was frozen in liquid nitrogen and stored at –40 °C.

## CPSF6 peptides

The peptides $CPSF6_{313-327}$ (CPSF6p) and $CPSF6_{313-327}$ with an extra cysteine at the C-terminus (CPSF6p-Cys) were synthetised by GenScript. Peptides were dissolved in water at a concentration of 2.5 mM and stored in aliquots at –40 °C.

## Labelling of CPSF6 peptide

CPSF6p-Cys was labelled with Alexa-Fluor 568-C5-maleimide (Thermo Fisher Scientific) added at an equimolar ratio in HEPES buffer pH 8. Labelling was verified by thin layer chromatography (TLC). No unconjugated dye was observed on TLC. CPSF6p-AF568 solution stored in aliquots at –40 °C.

## Single-molecule TIRF uncoating assay

Single-molecule imaging of viral particles was carried out using TIRF microscopy with microfluidic sample delivery according to our previously published methods (*Márquez et al., 2019*). Briefly, biotinylated viral particles were captured onto coverslips attached to PDMS microfluidic flow cells and imaged using a custom-built TIRF microscope with an ASI-RAMM frame (Applied Scientific Instrumentation), a Nikon 100×CFI Apochromat TIRF (1.49 NA) oil immersion objective and NicoLase laser system (*Nicovich et al., 2017*). Immobilised virions were treated with imaging buffer containing 200 nM pore forming protein (DLY or SLO) to permeabilise viral membrane and AF568-labelled CypA (0.8 µM) to paint the capsid. Drugs (LEN, PF74, BI-2) were added to the imaging buffer as stock solutions in DMSO (final concentration not exceeding 0.5%). Images were acquired at a rate of 1 frame per 6 s for 30 min unless specified otherwise.

## TIRF image analysis

Single-virion fluorescence traces were extracted from the TIRF image stacks using the JIM Immobilized Microscopy analysis package (*Walsh, 2021*, freely available on GitHub) and further analysed in MATLAB (The MathWorks Inc).

## Capsid opening via GFP release

Change-point analysis of GFP intensity traces using an algorithm (*Taylor, 2000*) implemented in C++ was used to identify the presence and time of steps corresponding to membrane permeabilisation and capsid opening. Step probabilities were calculated non-parametrically using 10,000 bootstrap iterations. Step times were calculated using least squares fitting. Traces were automatically sorted into four classes based on the following criteria: (1) loss of entire GFP signal in one step; (2) loss of GFP intensity in one large step (permeabilisation) and one small step (capsid opening, identified with ≥75% step probability and ≥75% signal loss); (3) loss of the majority of the GFP signal in one step with residual GFP signal persisting for the rest of the experiment; (4) no or insufficient GFP signal drop or traces with more than two steps (excluded from analysis). Capsid opening times were calculated for traces in class two as the time difference between permeabilisation and capsid opening. Survival curves were constructed from the pooled opening times acquired in independent uncoating experiments.

## Analysis of CA lattice stability via CypA paint

Heatmaps and median traces of leaky or closed particles were generated after aligning at traces in the corresponding category at the time of membrane permeabilisation. Traces of opening particles were aligned at the time of membrane permeabilisation (shown in the first panel) and aligned at the time of capsid opening (shown in the second panel).

## Quantification of bound molecules

The number of bound AF568-labelled CPSF6p molecules was determined by dividing the CPSF6p-AF568 fluorescence intensity associated with each capsid by the fluorescence intensity of a single CPSF6p-AF568 molecule. The fluorescence intensity of the single fluorophore was determined from the quantal photobleaching step in photobleaching traces of CPSF6p-AF568 molecules adsorbed sparsely to the coverslip surface and imaged continuously.

## Calculation of occupancy

The occupancy at equilibrium ($\theta_{eq}$) of FG binding pockets on the capsid with LEN was calculated using the equation $\theta_{eq} = c_{LEN} / (c_{LEN} + K_D)$, where $c_{LEN}$ is the LEN concentration and $K_D$ is the dissociation constant of the LEN-capsid interaction. The occupancy as a function of time ($\theta(t)$) was calculated using the following equation: $\theta(t) = \theta_{eq} 1-\exp(-(k_{on} \times c_{LEN} + k_{off}) \times t)$, where $\theta_{eq}$ is the occupancy at equilibrium, $c_{LEN}$ is the LEN concentration, $k_{on}$ is the association rate constant and $k_{off}$ is the dissociation rate constant. Estimates for $K_D$, $k_{on}$ and $k_{off}$ were obtained from TIRF imaging in this work ($K_D$ = 0.51 nM, $k_{on}$ = 1.84E5 $M^{-1}s^{-1}$ and $k_{off}$ = 1.84E-4 $s^{-1}$) or from published values obtained by surface plasmon resonance spectroscopy ($K_D$ = 0.24 nM, $k_{on}$ = 6.5E4 $M^{-1}s^{-1}$ and $k_{off}$ = 1.4E-5 $s^{-1}$) (*Link et al., 2020*).

## Production of pseudotyped-virus particles for cell-based assays

VSV-G pseudotyped GFP-encoding virus particles were generated by co-transfecting HEK293T cells with pCRV1-GagPol, pCSGW and pMD2.G (1:1.1:1.3 mol/mol) using PEI 25 K. The culture medium was removed 16 hr post-transfection and replenished with a fresh medium containing 10 mM $MgCl_2$, 0.5 mM $CaCl_2$ and 100 U of DNase. Virus-containing medium was harvested 72 hr post-transfection, centrifuged (2100 x *g*, 20 min, 4 °C) to remove cell debris, divided into aliquots and stored at –80 °C.

## Infection assays

Infection assays were performed in 96 well plates using $0.75 \times 10^5$ Jurkat cells per well pre-treated with the indicated drug (LEN, PF74) concentrations for 30 min at 37 °C. The cells were then infected in triplicate with VSV-G-pseudotyped GFP-encoding virus in the presence of polybrene (10 µg/ml; Sigma) at room temperature for 20 min, followed by spinoculation at 800 x *g* for 1 hr at room temperature. The culture media were removed and replenished with fresh media containing drugs at the indicated concentrations. At 48 hours post infection, the cells were fixed in 2% paraformaldehyde (Electron Microscopy Sciences) for 1 hr at room temperature and analyzsd by flow cytometry using a LSRFortessa cell analyzer (BD Biosciences) and FlowJo software. For experiments involving preincubation with drugs, the virus was incubated with LEN at the indicated concentrations for 48 hr at room temperature and then used to infect cells as above. We chose 48 hr because this time is sufficient for

binding at concentrations ≥50 pM to reach ≥95% of the respective equilibrium level (*Figure 4—figure supplement 1A*). For quantitative PCR (qPCR) analysis, virus was treated with DNase for 30 min at 37 °C and then used to infect $1\times10^5$ Jurkat cells in duplicate as described above. Cells were harvested 24 hours post infection and processed for qPCR.

### Quantitative PCR analysis
Genomic DNA was isolated from pelleted cells using a DNeasy blood and tissue kit (Qiagen). The concentration of purified DNA was determined using a Nanodrop spectrophotometer. To quantify total viral DNA, quantitative PCR was performed using sequence-specific primers and SsoAdvanced Universal SYBR Green Supermix (Bio-Rad). PCR conditions for vDNA amplification: Initial denaturation, 95 °C, 3 min; Denaturation, 95 °C, 10 s; Annealing/Extension, 60 °C, 30 s; 45 cycles.

### Expression and purification of CA for in vitro assembly experiments
*E. coli* C41 cells expressing CA were lysed and cell debris was removed by centrifugation. CA was precipitated by addition of 25% ammonium sulphate to the supernatant, collected by centrifugation, resuspended, and dialysed against 50 mM MES (pH 6.0), 20 mM NaCl, 1 mM DTT. The CA protein was further purified on a cation exchange column with a gradient from 20 mM to 1 M NaCl followed by size exclusion chromatography with Tris pH 8.0, 20 mM NaCl, 1 mM DTT and finally snap frozen.

### Turbidity assay to measure CA assembly kinetics
CA proteins were dialysed against 50 mM MES (pH 6.0), 1 mM DTT. CA proteins were assembled at a final concentration of 75 µM in the presence of 2% DMSO. LEN titration: LEN (final concentration between 0.5–50 µM) was added to the CA solution and assembly was initiated by adding IP6 (final concentration of 200 µM). IP6 titration: IP6 (final concentration between 50 and 150 µM) ±LEN (final concentration 50 µM) was added to the CA solution to induce assembly. The increase in $Abs_{350}$ was measured with a PHERAstar FSX Plate reader (BMG Labtech) in 384-well plate every 22 s with shaking after each measurement.

### Negative staining EM of self-assembled CA structures
The samples from the turbidity assay were allowed to sediment overnight. Then 5 µL of each sample was applied to a carbon coated grid (Cu, 400 mesh, Electron Microscopy Services) previously cleaned by glow discharge. The grids were then washed, and samples stained with 2% uranyl-acetate. Micrographs were taken at room temperature on a Tecnai Spirit (FEI) operated at an accelerated voltage of 120 keV and recorded with a Gatan 2k × 2k CCD camera. Images were collected with a total dose of ~30 e$^-$/$^2$ and a defocus of 1–3 µm.

### HIV particle production for cryo-electron tomography
Replication deficient VSV-G pseudotyped HIV-1 virions were produced in HEK293T cells using pCRV1-GagPol, pCSGW and pMD2.G as described previously (*Price et al., 2014*). At 24–48 hr post transfection, the supernatants were harvested and passed through 0.22 µm nitrocellulose filter. The virions were concentrated by ultracentrifugation through a 20% (w/v) sucrose cushion (2 hr at 28,000 rpm in a SW32 rotor [Beckman Coulter Life Sciences]). The pellet was resuspended in PBS, snap-frozen and stored at –80 °C. LEN-treated virions were incubated in presence of 700 nM LEN for 1.5 hr at room temperature prior to plunge-freezing for cryo-ET.

### Cryo-electron tomography of LEN-treated HIV particles
Colloidal gold beads (10 nm diameter) were added to the purified HIV particles and 6 µl of this suspension was applied to a C-Flat 2/2 3 C grid cleaned by glow discharge (20 mA, 40 s). The grids were blotted and plunge-frozen in liquid ethane using an FEI Vitrobot Mark II at 16 °C and 100% humidity. Tomographic tilt series between –40° and +40° with increments of 3°, defoci between –3 µm and –6 µm at a magnification of ×50,000 were acquired using Serial-EM (*Mastronarde, 2005*) on a TF2 Tecnai F20 transmission electron microscope under low-dose conditions at 200 kV and images recorded with a Falcon III direct electron detector. The Imod package (IMOD Version 4.9.0) was used to generate tomograms (*Kremer et al., 1996*). The alignment of 2D projection images of the tilt series was done using gold particles as fiducial markers. A 3D reconstruction was generated using back

projection of the tilt-series. All the morphological classes, including the double-layered capsids, were classified by visual inspection.

## Acknowledgements

This work was supported by NHMRC Ideas Grant APP1182212 (DAJ, TB), Wellcome Trust Collaborator Award 214344/Z/18/Z (GJT, LJ, DAJ, TB), NHMRC Investigator Grant APP1194263 (MWP), Australian Research Council Grants DP160101874 and DP200102871 (MWP) and a UNSW Scientia Award (KMRF). CM received an Australian Government Research Training Program Scholarship. Infrastructure support from the NHMRC Independent Research Institutes Infrastructure Support Scheme and the Victorian State Government Operational Infrastructure Support Program to St. Vincent's Institute are gratefully acknowledged. MWP is an NHMRC Leadership Fellow. We thank Sara Lawrence (St. Vincent's Institute) for expression and purification of recombinant DLY, Claire Dickson and Prabhjeet Phalora for critical feedback on the manuscript.

## Additional information

### Funding

| Funder | Grant reference number | Author |
| --- | --- | --- |
| National Health and Medical Research Council | APP1182212 | David A Jacques<br>Till Böcking |
| Wellcome Trust | 214344/Z/18/Z | Greg J Towers<br>Leo C James<br>David A Jacques<br>Till Böcking |
| National Health and Medical Research Council | APP1194263 | Michael W Parker |
| Australian Research Council | DP160101874 | Michael W Parker |
| Australian Research Council | DP200102871 | Michael W Parker |

The funders had no role in study design, data collection and interpretation, or the decision to submit the work for publication.

### Author contributions

KM Rifat Faysal, Formal analysis, Investigation, Methodology, Writing – original draft; James C Walsh, Software, Formal analysis, Investigation, Methodology, Writing – review and editing; Nadine Renner, Vaibhav B Shah, Andrew J Tuckwell, Formal analysis, Investigation, Writing – review and editing; Chantal L Márquez, Formal analysis, Investigation, Methodology, Writing – review and editing; Michelle P Christie, Resources, Writing – review and editing; Michael W Parker, Resources, Supervision, Funding acquisition, Writing – review and editing; Stuart G Turville, Resources, Supervision, Writing – review and editing; Greg J Towers, Funding acquisition, Writing – review and editing; Leo C James, Supervision, Funding acquisition, Writing – review and editing; David A Jacques, Conceptualization, Supervision, Funding acquisition, Writing – original draft; Till Böcking, Conceptualization, Formal analysis, Supervision, Funding acquisition, Methodology, Writing – original draft

### Author ORCIDs

Michael W Parker http://orcid.org/0000-0002-3101-1138
Stuart G Turville http://orcid.org/0000-0003-1918-5343
Greg J Towers http://orcid.org/0000-0002-7707-0264
Leo C James http://orcid.org/0000-0003-2131-0334
David A Jacques http://orcid.org/0000-0002-6426-4510
Till Böcking http://orcid.org/0000-0003-1165-3122

Decision letter and Author response
Decision letter https://doi.org/10.7554/eLife.83605.sa1
Author response https://doi.org/10.7554/eLife.83605.sa2

## Additional files

### Supplementary files
• MDAR checklist

### Data availability
The image analysis software is available on GitHub (*Walsh, 2021*). The microscopy image stacks for *Figures 2–4, 7 and 8*, the source data for the dose-response curves in *Figures 5 and 7* and the source data for the CA assembly curves in *Figure 6* are available on Dryad.

The following dataset was generated:

| Author(s) | Year | Dataset title | Dataset URL | Database and Identifier |
|---|---|---|---|---|
| Walsh J, Böcking T | 2024 | Pharmacologic hyperstabilisation of the HIV-1 capsid lattice induces capsid failure | https://doi.org/10.5061/dryad.18931zd49 | Dryad Digital Repository, 10.5061/dryad.18931zd49 |

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
