## [Editor Report]

This important study substantially advances our understanding of the effects of small molecule inhibitors on the structural integrity and stability of the HIV-1 capsid. Rigorous biochemical assays and state-of-the-art microscopy provide compelling support for the conclusions. The work will be of broad interest.

---

## [Decision Letter]

**Decision letter after peer review:**

[Editors’ note: the authors submitted for reconsideration following the decision after peer review. What follows is the decision letter after the first round of review.]

Thank you for submitting the paper "Pharmacologic hyperstabilisation of the HIV-1 capsid lattice induces capsid failure" for consideration by *eLife*. Your article has been reviewed by 3 peer reviewers, and the evaluation has been overseen by a Reviewing Editor and a Senior Editor. The reviewers have opted to remain anonymous.

Comments to the Authors:

We are sorry to say that, after consultation with the reviewers, we have decided that this work will not be considered further for publication by *eLife*. We agree that the most interesting and novel aspect of the paper is the proposal that virus inhibition at low LEN concentrations occurs only when a sufficient number of drug molecules have bound the capsid. The reviewers are not convinced that the presented data are strongly supportive of this model.

New data that more strongly support the proposed model of "lethal hyperstabilization" are needed. The key here is the experimental validation of the occupancy estimates and a more precise determination of the "requisite threshold occupancy". Also important is additional validation of the predictive value of the model, for example by more extensive testing of the effects of pre-incubation at low drug concentrations, or of time-of-addition effects on infectivity. We understand that this will take significant experimentation, and in such a case *eLife* recommends a decision to Reject.

Consolidated assessment:

(1) Experiments in Figures2-4 establish that LEN elicits the same capsid effects as PF74, which was characterized in a previous paper. LEN effects are observed at lower drug concentrations than PF74, consistent with its tighter binding affinity. IP6 counteracts the effects of LEN, but only at low drug concentrations. The concentration dependence is explained in terms of a "requisite occupancy threshold," estimated to be 30% of the potential binding sites on the capsid, that needs to be reached before structural effects on the capsid are seen. Occupancy is estimated using a simple equilibrium binding model; at 500 nM LEN, the occupancy threshold is reached in <2 min, and only after ~20 h at IC50. These experiments underlie the proposed "lethal hyperstabilization" model, i.e., that viral inhibition at low LEN concentrations occurs many hours after viral entry because it takes this long to reach the occupancy threshold. The model is tested in experiments that compare inhibition profiles of virions that are preincubated with drug vs untreated virions (Figure 5).

The model is an attractive explanation for how a capsid-targeting drug apparently inhibits integration, but the reviewers are not convinced that the data are strongly supportive of the claimed mechanism. Specifically: (1) there is no evidence presented that LEN occupancy is accurately described by a simple equilibrium model, (2) preincubation experiments show only a modest 1.3-fold effect at low LEN concentrations, (3) preincubation at low LEN does not indicate a clear difference in capsid stabilization and (4) alternative models that can explain LEN potency are not ruled out.

(2) Figures6-7 examine the effects of LEN on in vitro capsid assembly and purified virions. These effects manifest at high concentrations and are largely confirmatory of previous studies. It needs to be clarified if the "extra" capsid lattices in Figure 7 are due to LEN-induced assembly after maturation has been completed, and thus not strictly a capsid assembly defect. If so, their relevance to infectivity needs to be established.

(3) Finally, Figures8-9 examine the capsid effects of lower potency drugs (PF74 and BI-2) and a CPSF6-derived peptide, which binds to the same site as LEN. The results confirm previous results on PF74 and further validate LEN's potency. Results with CPSF6 peptide are suggested to indicate stabilization at lower occupancy compared to LEN, which is an interesting finding. However, only peptide occupancy is measured directly.

*Reviewer #1 (Recommendations for the authors):*

Specific comments:

As shown by the authors in Figure 5, HIV infection shows sensitivity to Len below 1 nM concentrations. At these concentrations, the single virion analysis doesn't show any major sensitivity to Len. In short, infectivity is sensitive below 1nM, Single virion assay shows sensitivity at 50nM, and in vitro assembly at about 5uM. At 1nM, there is not sufficient occupancy to support the claimed mechanism suggested by the authors. It is therefore unclear, at what stage of the infection does Len work and if capsid stabilization plays a role in the drug's action.

Only a small fraction (1-10%) of released HIV virions are infectious, while the single virion assay observes all virions. The authors have done good work characterizing the single virion assay, however, there are many technical issues still unresolved. For example:

I couldn't find the fraction of GFP positive and CypA negative virions observed in their experiments, this number does inform on the efficiency of the maturation in the vector system they used to produce their particles.

The percentages of leaky virions measured from virions treated with 500nM of Len fluctuate significantly in between figures. Specifically, Figure 2E shows the percentage of leaky virions is the same between no drug and 500nM Len while Figure 3B shows a significantly larger fraction of leaky virions in the presence of 500nM Len. While this can be due to different batches of virions, it highlights that many of the observations are acutely sensitive to viral preparations.

*Reviewer #2 (Recommendations for the authors):*

1. It is surprising how much capsid breakage the authors observe in their single-molecule experiments in the presence of IP6 (Figure 4A-B). In this paper, they report only ~30% of capsids remain intact after 30 min in the presence of 100uM IP6. In their previous work (Mallery et al., 2018), using a similar single-molecule assay to track the loss of GFP as capsids break open, they estimated the half-life of intact capsids was 10 h in the presence of 100uM IP6. Is this related to a change in the method? Could the authors comment on the reproducibility of these experiments and the limitations of the assay?

2. It seems plausible that the concentration of DLY/SLO used to permeabilize the virus membrane (or PFO as used in Marquez et al., 2018) could impact capsid integrity. The authors should perform a titration with their pore-forming proteins and report whether lower concentrations reduce capsid leakiness at baseline or delay the time to capsid opening.

3. Related to points #1 and #2, several of the capsid survival curves presented in the paper are reported as recycled data (e.g. Figure 4 shows the uncoating data in the absence of IP6 from Figure 3; Figure 8 – Supplement 1 shows control and drug data from Marquez et al. 2018; and Figure 9 – Supplement 1 shows control data from Marquez et al. compared to CPSF6). If data are collected under different conditions, that is to say, collected using a different pore-forming peptide at the permeabilization step (such as with PFO in the case of Marquez et al), it would not be appropriate to compare the level of capsid breakage and then attribute differences to other variables (like IP6 or CPSF6).

4. To further support their findings, the authors should perform single-molecule experiments with CA mutants that are known to destabilize or stabilize (such as E45A as tested in Marquez et al. 2018) the capsid structure – these should shift the balance towards leaky or closed phenotypes, respectively.

5. Is GFP release or the CypA paint signal affected by LEN resistance-associated mutations such as Q67H^+^N74D or M66I? Do these mutations reduce the effects of LEN in these experiments?

6. Line 409: Given the aberrant structures shown from CryoET, and the fact that the authors observe rapid loss of GFP signal with LEN in their single-molecule assays, it is strange that the authors could not find examples of holes or capsid breakage in the tomograms of LEN-treated samples. If the authors generated reconstructions (as stated in the methods), it would be helpful to see what these aberrant structures look like in 3D, since it appears there could be breaks in some of the images depicted in Figure 7.

*Reviewer #3 (Recommendations for the authors):*

In this article, Faisal et al., use a combinatorial approach to look at the mechanisms of HIV-capsid inhibition by the highly potent drug Lenacepavir (LEN). The effects of LEN on capsid assembly are nicely demonstrated and suggest that LEN might be involved in capsid stabilization. This conclusion is corroborated by IP6-LEN competition assay in vitro and cryo-ET analysis of native virions. The cryo-ET analysis provides novel insights into the formation of 2-layer capsid sheets. The authors seem to elude to excess capsid in virions by fluorescence analysis. However, whether the second layer originated from unassembled capsids (as proposed by the authors) or by incorporating excess gag into virions is unclear, as no quantitation of EM-tomograms is reported. Additionally, the details of cryo-ET volumetric segmentation and quantification are missing. The description and interpretation of the data in the Results sections and the conclusions are inconsistent, and somewhat confusingly presented for the general non-expert audience.

The paper mainly focuses on the effects of LEN and other low-potent molecules PF74, BI-2, or a small peptide of the host-factor CPSF6 on the stability of pre-formed in virio capsids. All these molecules, including the host-factor CPSF6 bind the same interface on the capsid, albeit with a different structural mechanism. The authors embark on multiple experiments to show the concentration-dependent effects of LEN on capsid stability. These experiments make use of a previously reported HIV-GagiGFP construct developed by other groups (PMID: 17728233, not cited) and employ single molecule TIRF imaging to sensitively quantify capsid opening, while also visualizing lattice disassembly by CypA-paint in an in vitro assay developed by the authors' group. Throughout the manuscript, CypA-paint heatmaps are shown for averaged single virus traces (albeit with different intensity associated with colors in the figures), with numbers and overlaid graphs to show lattice stabilization/ destabilization phenotypes. However, data showing the main population distribution analysis reveals a different story. Presumably, capsids retaining (1) iGFP (closed) and (2) lost-iGFP but retaining CypA-paint (opening) are the stabilized population of closed or open cores containing lattices, respectively. The others (3) leaky, and (4) short-lived are cores completely losing the lattice and are examples of instability. Taking this into consideration, data in Figure 2D, 3A, and 4A (-IP6), show that capsids open much quicker with increasing concentrations of LEN, and the lattice becomes highly unstable (sum of open + closed cones) in Figure 2E, 3B and 4B (-IP6) (note lattice stabilization is lost with LEN > 5nM). The data thus shows opposite capsid de-stabilization effects by LEN and is thus inconsistent with the main conclusion that LEN induces 'lethal hyperstability' (line 507), and discordant with other cited reports (Bester et al). Quantification for the population of closed and open capsids in PF74, BI-2, and CPSF6-peptide is not shown. In these cases, only the iGFP-loss opening phenotype is plotted and control plots are extrapolated from a prior publication.

The existing literature suggests LEN 0.05-500nM, and PF74 10uM stabilize canonical virus strain pR9, and pNL capsids (Bester et al., PMID: 30393786, PMID: 27322072), and BI-2 to de-stabilize the same capsids (PMID: 27322072), using similar in vitro assays employed here and also in living cells. In my opinion, the disagreements appear to stem from the use of the HIV-iGFP virus system in the current work. The authors acknowledge the 'leaky' or unstable capsids in their system comprise the major population, which exhibits complete loss of iGFP and the lattice. A subpopulation of cores likely opened but with stabilized lattices, and only a few minor populations of presumably intact cores in their system. Concomitantly, they have also established previously that the major population of capsids derived from these iGFP viruses is deformed (aberrant cones and tubes) (PMID: 29877795), when compared with the major population of cones in mature particles of conventional pNL- or pR9- based virus systems. The unclear understanding of what relates to relevant classical cones found in infectious virions, in the iGFP-system is a caveat. Analysis of CypA-paint and proportions of capsids of native HIV particles (not iGFP-tagged) as shown in Figure 7C and Suppl. Figure 7, will be needed for a rigorous interpretation of their data.

Regarding capsid occupancy, the authors use previously determined binding constants (Kon, Koff) to model the occupancy of the drugs and CPSF6-peptide on the capsid, which is novel. Since occupancy is the main part of the story, their modelling should be experimentally verified. The pre-treatment experiments provide some light; however, the data interpretation is confusing. The total stabilized capsids (sum of closed and open) are the same for pre-treated cores, thus pre-treatment offers no particular advantage. These results do not validate their model.

It is unclear to me what is meant by occupancy, if not coating the whole capsids. Can they experimentally determine whether their occupancy modeling is correct? At least in the context of CPSF6-peptide and CypA, both of which bind with much lower affinity (uM range) to capsid, seem to show plateaued capsid occupancy curves within seconds at sub-KD concentrations (CPSF6-pep 5uM, and CypA 1uM) used in their experiments. Noted from Figure 9 Suppl. Figure 2, where a very nice titration experiment shows the binding of sub-KD concentrations of CPSF6-peptide. The difference in the plateau of fluorescence shows the different occupancy, but occupancy itself is complete within 20 sec (plateau), is it not the case? This data already shows that the lower concentration of CPSF6-peptides will not continue occupying additional capsid sites and show a linear curve. Contrastingly, in the models, LEN shows a slow rising curve extending days. At twice the KD 0.5 and 5 nM of LEN (KD ~230 pM, PMID: 36190128) there should be enough drug to occupy entire capsids, in the invitro assays. The authors should clarify and include the formula/calculations of the occupancy. Also, was the concentration of immobilized capsid itself in the invitro uncoating assay considered for occupancy modeling?

In summary, while I feel like the assembly effects are sufficiently supported, the disassembly and/or stabilizing effects need more rigor to be established. The Results section and discussion should be accordingly written and the fraction of CypA-paint puncta should be overlaid on GFP+ capsid plots for clarity.

---

## [Author Response]

[Editors’ note: The authors appealed the original decision. What follows is the authors’ response to the first round of review.]

Consolidated assessment:1) Experiments in Figures2-4 establish that LEN elicits the same capsid effects as PF74, which was characterized in a previous paper. LEN effects are observed at lower drug concentrations than PF74, consistent with its tighter binding affinity. IP6 counteracts the effects of LEN, but only at low drug concentrations. The concentration dependence is explained in terms of a "requisite occupancy threshold," estimated to be 30% of the potential binding sites on the capsid, that needs to be reached before structural effects on the capsid are seen. Occupancy is estimated using a simple equilibrium binding model; at 500 nM LEN, the occupancy threshold is reached in <2 min, and only after ~20 h at IC50. These experiments underlie the proposed "lethal hyperstabilization" model, i.e., that viral inhibition at low LEN concentrations occurs many hours after viral entry because it takes this long to reach the occupancy threshold. The model is tested in experiments that compare inhibition profiles of virions that are preincubated with drug vs untreated virions (Figure 5).The model is an attractive explanation for how a capsid-targeting drug apparently inhibits integration, but the reviewers are not convinced that the data are strongly supportive of the claimed mechanism. Specifically: (1) there is no evidence presented that LEN occupancy is accurately described by a simple equilibrium model, (2) preincubation experiments show only a modest 1.3-fold effect at low LEN concentrations, (3) preincubation at low LEN does not indicate a clear difference in capsid stabilization and (4) alternative models that can explain LEN potency are not ruled out.

Point (1) This is an important point. We have added a new experiment to measure LEN binding to authentic capsids as a function of time by displacement of labelled CPSF6 peptide as a “paint” probe for the FG binding pocket. This experiment directly relates occupancy to capsid rupture, and strengthens the lethal hyperstability mechanism that dominates in the mid-concentration range (≥0.5 nM).

Points (2) – (4) Our study does not investigate the inhibitory mechanism that dominates at concentrations close to the IC50 (~0.1 nM). Previous studies have shown inhibition in this regime occurs at the stage of integration, and we do not dispute these findings. Specifically, we do not suggest that lethal hyperstability explains inhibition at these low concentrations. To avoid confusion, we have rewritten the corresponding sections in the manuscript.

(2) Figures6-7 examine the effects of LEN on in vitro capsid assembly and purified virions. These effects manifest at high concentrations and are largely confirmatory of previous studies. It needs to be clarified if the "extra" capsid lattices in Figure 7 are due to LEN-induced assembly after maturation has been completed, and thus not strictly a capsid assembly defect. If so, their relevance to infectivity needs to be established.

The CA self-assembly experiment provides new insight by showing that IP6 and LEN synergise to promote CA assembly, but drive assembly of hexameric vs. conical lattices. This antagonism helps explain previous studies revealing aberrant capsid morphologies inside virions when LEN is added before virus maturation. Previous in vitro assembly experiments only showed that LEN accelerates CA assembly at high salt but did not test the opposing effects of IP6 and LEN on morphology we show here.

The cryoET analysis in Figure 7 indeed shows mature virions treated with LEN (as noted e.g. in the figure legend). The assembly of free CA (that is not already part of the mature capsid) into additional lattices confirms the overassembly phenotype observed in vitro, and we agree that LEN would already cause defects during maturation. We have condensed this section and moved this figure to the Supplement to keep the focus on the antagonism between IP6 and LEN that is mainly documented in the in vitro assembly experiments.

(3) Finally, Figures8-9 examine the capsid effects of lower potency drugs (PF74 and BI-2) and a CPSF6-derived peptide, which binds to the same site as LEN. The results confirm previous results on PF74 and further validate LEN's potency. Results with CPSF6 peptide are suggested to indicate stabilization at lower occupancy compared to LEN, which is an interesting finding. However, only peptide occupancy is measured directly.

The main point of this section is to show that different molecules binding to the FG binding pocket have different effects on capsid stability. The observed differences can be explained using existing structures (see discussion) and have implications for interpreting experiments that use these compounds as tools to modulate capsid uncoating. Further validation of occupancy for PF74 and BI-2 are the topic of future work.

Reviewer #1 (Recommendations for the authors):Specific comments:As shown by the authors in Figure 5, HIV infection shows sensitivity to Len below 1 nM concentrations. At these concentrations, the single virion analysis doesn't show any major sensitivity to Len. In short, infectivity is sensitive below 1nM, Single virion assay shows sensitivity at 50nM, and in vitro assembly at about 5uM. At 1nM, there is not sufficient occupancy to support the claimed mechanism suggested by the authors. It is therefore unclear, at what stage of the infection does Len work and if capsid stabilization plays a role in the drug's action.

The block to infection at LEN concentrations close to the EC50 (~0.1 nM) occurs at the stage of integration as shown in earlier work (PMID 33060363, PMID 32612233). In this manuscript, we describe the inhibitory mechanism (termed 'lethal hyperstabilisation') that becomes dominant at concentrations close to the EC95 (≥0.5 nM). Thus, we do not dispute the original findings, but add a mechanism that operates in a different concentration range. We have made this clearer by rewriting the introduction accordingly.

The data in our original manuscript showed that 5 nM LEN has an intermediate effect on capsid integrity when measured over 30 min (Figure 3). We argued that this is because drug binding is so slow such that the occupancy required for rapid breakage is not reached within the 30 min experiment. We used a pre-incubation experiment to show that low nM LEN can induce capsid breakage when sufficient time is allowed for drug binding to equilibrate. But we agree that we did not directly show a relationship between LEN binding and capsid breakage.

To fill this gap, we have added new long-term experiments in which we image LEN binding (via displacement of a fluorescent CPSF6 peptide) and capsid rupture (via GFP release). We observed that binding of LEN to IP6-stabilised cores causes the capsid to rupture (albeit very slowly) down to the sub-nM concentrations tested in the experiment. This shows that the lethal hyperstability mechanism is active at a concentration corresponding to the second inhibitory phase.

The new experiments have been added to the manuscript as Figure 4.

Only a small fraction (1-10%) of released HIV virions are infectious, while the single virion assay observes all virions. The authors have done good work characterizing the single virion assay, however, there are many technical issues still unresolved. For example:I couldn't find the fraction of GFP positive and CypA negative virions observed in their experiments, this number does inform on the efficiency of the maturation in the vector system they used to produce their particles.

Immature particles can indeed be easily identified. Unlike particles that have undergone proteolysis, immature particles do not release GFP, because it remains part of the Gag polyprotein anchored to the viral membrane. CypA binding to these particles is slow. Particles identified as immature on the basis of these criteria are excluded from analysis since they do not contain a mature capsid.

The fraction of immature particles varies between virus preparation and is between 4–12% (mean of 7%) for iGFP particles used for Figures 2 and 3. As expected, the fraction of immature particles does not depend on addition of LEN or IP6 added during the uncoating experiment. We have added this quantification and example TIRF traces of immature particles as the new Figure 2—figure supplement 1.

The percentages of leaky virions measured from virions treated with 500nM of Len fluctuate significantly in between figures. Specifically, Figure 2E shows the percentage of leaky virions is the same between no drug and 500nM Len while Figure 3B shows a significantly larger fraction of leaky virions in the presence of 500nM Len. While this can be due to different batches of virions, it highlights that many of the observations are acutely sensitive to viral preparations.

We have added a bar chart of the fraction of "leaky" capsids from independent TIRF uncoating experiments in absence and presence of LEN to show the variability within each condition and compare between conditions (Figure 3—figure supplement 3). This analysis shows that the leaky fraction of experiments in Figures 2 and 3 is 56±5% for control virions. Addition of 500 nM LEN during the uncoating experiment causes an increase in the fraction of particles that immediately release GFP (65±7%), which we attribute to rapid LEN-induced capsid rupture (too fast to be resolved as a separate step in the uncoating traces recorded with a frame rate of 1 frame every 6 s).

Reviewer #2 (Recommendations for the authors):1. It is surprising how much capsid breakage the authors observe in their single-molecule experiments in the presence of IP6 (Figure 4A-B). In this paper, they report only ~30% of capsids remain intact after 30 min in the presence of 100uM IP6. In their previous work (Mallery et al., 2018), using a similar single-molecule assay to track the loss of GFP as capsids break open, they estimated the half-life of intact capsids was 10 h in the presence of 100uM IP6. Is this related to a change in the method? Could the authors comment on the reproducibility of these experiments and the limitations of the assay?

The analysis of capsid stability ±IP6 in *Mallery et al.* focuses on the subset of assembled capsids (i.e. the plot excludes leaky capsids). In contrast, Figures 2–5 of our current manuscript show survival curves of all capsids, including the "leaky" fraction (as indicated by the y-axis label). As a result, the scale of the y-axis is different in these two papers.

This is confusing for the reader and we thank the referee for bringing this to our attention. We have reverted y-axis of survival plots to the previous format that excludes the leaky fraction to allow direct comparison with previous papers and provide separate analysis of the leaky fraction in Figure 3—figure supplement 3.

IP6-mediated stabilisation is reproducible as shown in six previous papers (see PMID 29848441, PMID 31851928, PMID 33524070, PMID 33692109, PMID 36624347, PMID 36289397), whereby we find that maximum stabilisation requires fresh IP6 solutions.

Limitations of the assay are explained in the legend of a new Figure 2—figure supplement 2.

2. It seems plausible that the concentration of DLY/SLO used to permeabilize the virus membrane (or PFO as used in Marquez et al., 2018) could impact capsid integrity. The authors should perform a titration with their pore-forming proteins and report whether lower concentrations reduce capsid leakiness at baseline or delay the time to capsid opening.

We have conducted a titration of SLO, which shows that the pore-forming protein does not affect capsid stability (added as Figure 2—figure supplement 3).

3. Related to points #1 and #2, several of the capsid survival curves presented in the paper are reported as recycled data (e.g. Figure 4 shows the uncoating data in the absence of IP6 from Figure 3; Figure 8 – Supplement 1 shows control and drug data from Marquez et al. 2018; and Figure 9 – Supplement 1 shows control data from Marquez et al. compared to CPSF6). If data are collected under different conditions, that is to say, collected using a different pore-forming peptide at the permeabilization step (such as with PFO in the case of Marquez et al), it would not be appropriate to compare the level of capsid breakage and then attribute differences to other variables (like IP6 or CPSF6).

All survival curves in a figure, including those curves that have been replotted to facilitate comparison with new treatments, were collected at the same time and with identical experimental conditions. Indeed, the reason for replotting these curves is that they constitute the appropriate control for the corresponding set of experiments.

4. To further support their findings, the authors should perform single-molecule experiments with CA mutants that are known to destabilize or stabilize (such as E45A as tested in Marquez et al. 2018) the capsid structure – these should shift the balance towards leaky or closed phenotypes, respectively.

We have added an experiment with a new protocol that includes an equilibration period during which broken and highly unstable capsids that do not respond to IP6 are allowed to decay away. We can then test the effect of LEN on the remaining IP6-responsive (functionally relevant) subset of capsids. This avoids the complexity of convolving intrinsic instability with drug-induced breakage, thus removing the need to include extensive controls that account for variability in the unstable (leaky/short-lived) fraction between viral preparations. The extensive analysis of LEN occupancy on capsid integrity from this new approach is presented in Figure 4.

We agree that testing mutants that modulate capsid stability and/or LEN binding is an interesting future direction to further dissect the interplay between intrinsic stability and LEN-induced hyperstability. In our opinion, this is outside the scope of this (data-rich) paper that provides strong evidence for drug occupancy-dependent capsid breakage of wild type capsids.

5. Is GFP release or the CypA paint signal affected by LEN resistance-associated mutations such as Q67H^+^N74D or M66I? Do these mutations reduce the effects of LEN in these experiments?

We have not tested these mutants, see response to Reviewer #2 Point 4.

6. Line 409: Given the aberrant structures shown from CryoET, and the fact that the authors observe rapid loss of GFP signal with LEN in their single-molecule assays, it is strange that the authors could not find examples of holes or capsid breakage in the tomograms of LEN-treated samples. If the authors generated reconstructions (as stated in the methods), it would be helpful to see what these aberrant structures look like in 3D, since it appears there could be breaks in some of the images depicted in Figure 7.

At the resolution of our tomograms we are not confident in assigning apparent discontinuities as defined holes in the capsid.

Reviewer #3 (Recommendations for the authors):In this article, Faisal et al., use a combinatorial approach to look at the mechanisms of HIV-capsid inhibition by the highly potent drug Lenacepavir (LEN). The effects of LEN on capsid assembly are nicely demonstrated and suggest that LEN might be involved in capsid stabilization. This conclusion is corroborated by IP6-LEN competition assay in vitro and cryo-ET analysis of native virions. The cryo-ET analysis provides novel insights into the formation of 2-layer capsid sheets. The authors seem to elude to excess capsid in virions by fluorescence analysis. However, whether the second layer originated from unassembled capsids (as proposed by the authors) or by incorporating excess gag into virions is unclear, as no quantitation of EM-tomograms is reported. Additionally, the details of cryo-ET volumetric segmentation and quantification are missing. The description and interpretation of the data in the Results sections and the conclusions are inconsistent, and somewhat confusingly presented for the general non-expert audience.The paper mainly focuses on the effects of LEN and other low-potent molecules PF74, BI-2, or a small peptide of the host-factor CPSF6 on the stability of pre-formed in virio capsids. All these molecules, including the host-factor CPSF6 bind the same interface on the capsid, albeit with a different structural mechanism. The authors embark on multiple experiments to show the concentration-dependent effects of LEN on capsid stability. These experiments make use of a previously reported HIV-GagiGFP construct developed by other groups (PMID: 17728233, not cited) and employ single molecule TIRF imaging to sensitively quantify capsid opening, while also visualizing lattice disassembly by CypA-paint in an in vitro assay developed by the authors' group. Throughout the manuscript, CypA-paint heatmaps are shown for averaged single virus traces (albeit with different intensity associated with colors in the figures), with numbers and overlaid graphs to show lattice stabilization/ destabilization phenotypes. However, data showing the main population distribution analysis reveals a different story. Presumably, capsids retaining (1) iGFP (closed) and (2) lost-iGFP but retaining CypA-paint (opening) are the stabilized population of closed or open cores containing lattices, respectively. The others (3) leaky, and (4) short-lived are cores completely losing the lattice and are examples of instability. Taking this into consideration, data in Figure 2D, 3A, and 4A (-IP6), show that capsids open much quicker with increasing concentrations of LEN, and the lattice becomes highly unstable (sum of open + closed cones) in Figure 2E, 3B and 4B (-IP6) (note lattice stabilization is lost with LEN > 5nM). The data thus shows opposite capsid de-stabilization effects by LEN and is thus inconsistent with the main conclusion that LEN induces 'lethal hyperstability' (line 507), and discordant with other cited reports (Bester et al). Quantification for the population of closed and open capsids in PF74, BI-2, and CPSF6-peptide is not shown. In these cases, only the iGFP-loss opening phenotype is plotted and control plots are extrapolated from a prior publication.

As noted above, our original manuscript did not explain our results clearly enough, which might have led to some confusion about our message.

The ‘lethal hyperstability’ mechanism proposes opposing effects of LEN on the low-curvature (hexameric) part of the capsid and the high-curvature (pentamer-rich) part of the capsid. LEN hyperstabilises the former at the expense of the latter, converting a closed cone to a highly stabilised CA lattice with open edges.

Bester et al. show that CA puncta persist for longer in cells treated with LEN but do not determine whether the cone is still intact or whether it has ruptured. To distinguish between these possibilities, we use a marker for capsid integrity (encapsidated GFP as a content marker) in combination with a reporter for the number of CA molecules in the lattice (CypA paint). Our data showing hyperstabilisation of CA lattices with LEN (CypA paint read-out) is consistent with the data by Bester et al., but we additionally show that these lattices no longer form a closed cone (GFP content marker release).

We do not suggest that ‘lethal hyperstability’ is dominant at IC50 and we do not dispute the LEN mechanisms that have been suggested to operate at lower concentrations (IC50) by Bester et al. (and other papers). Instead our paper specifically investigates the LEN mechanism at concentrations around the IC95.

Changes to the paper:

(1) We have revised our manuscript by removing unnecessary complexity in our analysis/writing and by adding new experiments with improved experimental design.

(2) We have rewritten the introduction to make this point clearer from the outset.

The existing literature suggests LEN 0.05-500nM, and PF74 10uM stabilize canonical virus strain pR9, and pNL capsids (Bester et al., PMID: 30393786, PMID: 27322072), and BI-2 to de-stabilize the same capsids (PMID: 27322072), using similar in vitro assays employed here and also in living cells. In my opinion, the disagreements appear to stem from the use of the HIV-iGFP virus system in the current work. The authors acknowledge the 'leaky' or unstable capsids in their system comprise the major population, which exhibits complete loss of iGFP and the lattice. A subpopulation of cores likely opened but with stabilized lattices, and only a few minor populations of presumably intact cores in their system. Concomitantly, they have also established previously that the major population of capsids derived from these iGFP viruses is deformed (aberrant cones and tubes) (PMID: 29877795), when compared with the major population of cones in mature particles of conventional pNL- or pR9- based virus systems. The unclear understanding of what relates to relevant classical cones found in infectious virions, in the iGFP-system is a caveat. Analysis of CypA-paint and proportions of capsids of native HIV particles (not iGFP-tagged) as shown in Figure 7C and Suppl. Figure 7, will be needed for a rigorous interpretation of their data.

These papers (Bester et al., PMID: 30393786, PMID: 27322072) do not include a content marker and thus do not distinguish between stabilised closed cones and stabilised open lattices. The lattice stabilisation observed in those papers is consistent with our CypA paint results for LEN and PF74. Using the content marker we additionally show that these stabilised lattices are no longer closed cones.

As such, our observations extend those of previous papers rather than disagree with them. As outlined in the following paragraphs, we further note that (A) the HIV-iGFP virus system is well established and characterised in the context of capsid uncoating; (B) the presence different assembly states (incomplete vs intact) is similar to HIV without iGFP; (C) single-molecule analysis allows these subsets to be distinguished from each other such that we can resolve the effects of molecules (drugs, IP6) on the properties of intact capsids.

(A) The iGFP construct has been validated by different groups using live cell imaging studies to determine when and where the capsid first opens (e.g. PMID 28784755, PMID 33649225, DOI 10.1101/2023.08.22.553958 [bioRxiv preprint using iYFP]). These studies show that viruses with Gag-internal fluorescent protein lead to productive infection with normal kinetics, consistent with the formation of a proper capsid.

(B) As noted by the referee and consistent with our previous work (PMID 29877795), ~60% of iGFP virus particles contain "leaky" capsids that do not retain GFP. This observation is consistent with cryo-electron tomography analysis of 107 intact HIV particles showing that "most of the cores showed one or more local regions where the CA lattice was disrupted or absent" (PMID 27980210). Thus, the capsid assembly state of iGFP particles appears to be similar to what is observed in HIV particles without iGFP.

(C) The power of our single-molecule analysis is that we can distinguish leaky/short-lived capsid from properly formed (long-lived) capsids. We can then quantify the effects of LEN on these subsets relative to control conditions. That is to say, we can analyse the effect of compounds on capsid stability, regardless of whether leaky capsids represent a large subset.

To make this clearer, we have changed the experimental design in the experiments added in revision. We permeabilized virions in the presence of IP6 and waited for IP6-insensitive (leaky/unstable) capsids to decay before testing the binding and effects of LEN on the subset of IP6-stabilised cores, see Figure 4.In repsonse to the suggestion for additional experiments with dark HIV and CypA paint: We have previously shown that CypA paint analysis of the subset of properly formed capsids in dark HIV particles (produced using pCRV1-GagPol and pCSGW) exhibit essentially the same uncoating kinetics (survival curves) as iGFP particles, with IP6 extending the half-life of closed capsids from minutes to hours (PMID 33692109, PMID 31851928, PMID 33524070, PMID 36624347). As explained above, the fraction of leaky capsids in dark particles cannot be measured with CypA paint alone.

To make this clearer, we have changed the experimental design in the experiments added in revision. We permeabilized virions in the presence of IP6 and waited for IP6-insensitive (leaky/unstable) capsids to decay before testing the binding and effects of LEN on the subset of IP6-stabilised cores, see Figure 4.We have previously shown that CypA paint analysis of the subset of properly formed capsids in dark HIV particles (produced using pCRV1-GagPol and pCSGW) exhibit essentially the same uncoating kinetics (survival curves) as iGFP particles, with IP6 extending the half-life of closed capsids from minutes to hours (PMID 33692109, PMID 31851928, PMID 33524070, PMID 36624347). As explained above, the fraction of leaky capsids in dark particles cannot be measured with CypA paint alone.

Existing literature: These papers (Bester et al., PMID: 30393786, PMID: 27322072) do not include a content marker and thus do not distinguish between stabilised closed cones and stabilised open lattices. The lattice stabilisation observed in those papers is consistent with our CypA paint results for LEN and PF74. Using the content marker we additionally show that these stabilised lattices are no longer closed cones.

Regarding capsid occupancy, the authors use previously determined binding constants (Kon, Koff) to model the occupancy of the drugs and CPSF6-peptide on the capsid, which is novel. Since occupancy is the main part of the story, their modelling should be experimentally verified. The pre-treatment experiments provide some light; however, the data interpretation is confusing. The total stabilized capsids (sum of closed and open) are the same for pre-treated cores, thus pre-treatment offers no particular advantage. These results do not validate their model.

Great suggestion – we have measured the LEN binding kinetics on IP6-stabilised cores and added a new section describing this experiment to the revised manuscript (Figure 4).

It is unclear to me what is meant by occupancy, if not coating the whole capsids. Can they experimentally determine whether their occupancy modeling is correct? At least in the context of CPSF6-peptide and CypA, both of which bind with much lower affinity (uM range) to capsid, seem to show plateaued capsid occupancy curves within seconds at sub-KD concentrations (CPSF6-pep 5uM, and CypA 1uM) used in their experiments. Noted from Figure 9 Suppl. Figure 2, where a very nice titration experiment shows the binding of sub-KD concentrations of CPSF6-peptide. The difference in the plateau of fluorescence shows the different occupancy, but occupancy itself is complete within 20 sec (plateau), is it not the case? This data already shows that the lower concentration of CPSF6-peptides will not continue occupying additional capsid sites and show a linear curve. Contrastingly, in the models, LEN shows a slow rising curve extending days. At twice the KD 0.5 and 5 nM of LEN (KD ~230 pM, PMID: 36190128) there should be enough drug to occupy entire capsids, in the invitro assays. The authors should clarify and include the formula/calculations of the occupancy. Also, was the concentration of immobilized capsid itself in the invitro uncoating assay considered for occupancy modeling?

(i) Occupancy = fraction of FG binding pockets on the capsid that are occupied with a LEN molecule. Definition added to the legend of Figure 4—figure supplement 1. (ii) It is correct that CPSF6-peptide binding reaches its equilibrium quickly because the on- and off-rates are high (in contrast to LEN binding). (iii) We have added the formula to calculate occupancy at equilibrium to the legend of Figure 5 and the formula to calculate occupancy as a function of time to the legend of Figure 4–Figure Supplement 1.

In summary, while I feel like the assembly effects are sufficiently supported, the disassembly and/or stabilizing effects need more rigor to be established. The Results section and discussion should be accordingly written and the fraction of CypA-paint puncta should be overlaid on GFP+ capsid plots for clarity.

We have added new experiments to fully support the drug mechanism that operates in the low nM concentration range.